# Wild-type bone marrow cells repopulate tissue resident macrophages and reverse the impacts of homozygous CSF1R mutation

Dylan Carter-Cusack[1]*, Stephen Huang[1], Sahar Keshvari[1], Omkar Patkar[1], Anuj Sehgal[1], Rachel Allavena[2], Robert A. J. Byrne[3], B. Paul Morgan[3], Stephen J. Bush[4], Kim M. Summers[1], Katharine M. Irvine[1‡], David A. Hume[1‡*]

**1** Mater Research Institute-University of Queensland, Translational Research Institute, Woolloongabba, Brisbane, Australia, **2** School of Veterinary Science, The University of Queensland, Gatton, Australia, **3** UK Dementia Research Institute Cardiff, School of Medicine, Cardiff University, Cardiff, United Kingdom, **4** School of Automation Science and Engineering, Xi'an Jiaotong University, Xi'an, Shaanxi, China

‡ These authors are joint senior authors on this work.
* david.hume@uq.edu.au (DAH); d.cartercusack@student.uq.edu.au (DCC)

## Abstract

Adaptation to existence outside the womb is a key event in the life of a mammal. The absence of macrophages in rats with a homozygous mutation in the colony-stimulating factor 1 receptor (*Csf1r*) gene (*Csf1rko*) severely compromises pre-weaning somatic growth and maturation of organ function. Transfer of wild-type bone marrow cells (BMT) at weaning rescues tissue macrophage populations permitting normal development and long-term survival. To dissect the phenotype and function of macrophages in postnatal development, we generated transcriptomic profiles of all major organs of wild-type and *Csf1rko* rats at weaning and in selected organs following rescue by BMT. The transcriptomic profiles revealed subtle effects of macrophage deficiency on development of all major organs. Network analysis revealed a common signature of CSF1R-dependent resident tissue macrophages that includes the components of complement C1Q (*C1qa-/b/c* genes). Circulating C1Q was almost undetectable in *Csf1rko* rats and rapidly restored to normal levels following BMT. Tissue-specific macrophage signatures were also identified, notably including sinus macrophage populations in the lymph nodes. Their loss in *Csf1rko* rats was confirmed by immunohistochemical localisation of CD209B (SIGNR1). By 6-12 weeks, *Csf1rko* rats succumb to emphysema-like pathology associated with the selective loss of interstitial macrophages and granulocytosis. This pathology was reversed by BMT. Along with physiological rescue, BMT precisely regenerated the abundance and expression profiles of resident macrophages. The exception was the brain, where BM-derived microglia-like cells had a distinct expression profile compared to resident microglia. In addition, the transferred BM failed to restore blood monocyte or CSF1R-positive bone marrow progenitors. These studies provide a model for the pathology and treatment of CSF1R mutations in humans and the innate immune deficiency associated with prematurity.

**Data availability statement:** The raw sequencing files, in the form of .fastq files, are deposited in the European Nucleotide Archive under the study accession number PRJEB76985. The entire processed RNA-Seq dataset is also available for download as an excel spreadsheet alongside the supplementary materials at the following location (https://doi.org/10.48610/6d347aa). A web app has also been created to provide an accessible method for visualising the transcriptomic atlas (https://github.com/dylancartercusack/RNA-Seq-Web-App).

**Funding:** The original generation of the Csf1rko rat was supported by Medical Research Council (UK) grant MR/M019969/1 to DAH. DAH is supported by NHMRC (Australia) Investigator Grant 2009750. This work was funded by an Australian Research Council grant DP210102998 to DAH. DCC was supported by a UQ Postgraduate Scholarship and Mater Research Frank Clair Scholarship. The funders had no role in study design, data collection and analysis, decision to publish, or preparation of the manuscript.

**Competing interests:** The authors have declared that no competing interests exist.

## Author summary

Macrophages are essential immune cells that play critical roles in tissue development, maintenance, and repair. They are present in all organs of the body, contributing to organ-specific functions and systemic homeostasis. The growth and activity of macrophages depend on signals from the colony-stimulating factor 1 receptor (CSF1R). Mutations in the CSF1R gene disrupt macrophage function, leading to severe developmental abnormalities. In this study, we investigated the consequences of macrophage absence in rats with a homozygous mutation in *Csf1r* (*Csf1rko*). We generated transcriptomic profiles of all major organs of WT and *Csf1rko* rats, and discovered that these animals exhibit delayed organ maturation, impaired growth and early mortality. We also identified unique molecular signatures of macrophages globally as well as in specific organs, and uncovered their critical role in maintaining tissue integrity and function. By transplanting wild-type bone marrow cells at weaning, we restored macrophage populations in most tissues, reversing the growth defects and rescuing organ function.

## Introduction

Infectious disease and failure to thrive in the early postnatal period remain major challenges to both neonatal survival and resilience. The transition from intra-uterine to postnatal life requires immediate adaptation of the immune system to deal with pathogen challenge including the expansion and tissue-specific adaptation of tissue macrophage populations [1,2]. Aside from their role in innate immunity, macrophages are believed to be required for numerous developmental processes, including angiogenesis, adipogenesis, branching morphogenesis and neuronal patterning [3–6]. In terms of developmental maturity, neonatal rodents approximate preterm births in humans [7]. Preterm and low birth weight human infants are at especially high risk of developing infections and organ damage associated with persistently high mortality and morbidity, and lifelong health consequences for survivors. Susceptibility to infection in these infants is attributed in large part to the immaturity of both innate and acquired immune systems [8,9].

The differentiation, development and survival of resident tissue macrophages is controlled by signals from the macrophage colony-stimulating factor receptor (CSF1R) and its two ligands, CSF1 and IL34 [10]. Homozygous recessive mutations in the human *CSF1R* gene lead to severe postnatal developmental abnormalities [11,12]. Homozygous *Csf1r* mutation in inbred FVB/J mice produces pre-weaning lethality [13], which can be rescued in around 50% of pups by neonatal bone marrow cell transfer [14]. To provide an alternative model for CSF1R deficiency syndrome, and to dissect the functions of macrophages in development, we mutated the *Csf1r* gene in the rat [15]. Despite the loss of the abundant macrophage populations in the embryo, which are ablated in *Csf1rko* embryos, *Csf1rko* rats are almost indistinguishable macroscopically and histologically from littermates throughout embryonic development and at birth [16]. However, after birth the *Csf1rko* pups fail to thrive. The *Csf1rko* impacts somatic growth and organ development and limits the lifespan to 8–12 weeks [15,17]. Remarkably, the *Csf1rko* phenotype in rats can be reversed by intraperitoneal adoptive transfer of wild-type bone marrow cells (BMT) at weaning (3 weeks) without conditioning. Repopulation of resident tissue macrophage populations with donor-derived cells leads to restoration of somatic growth, long-term survival and even male and female fertility [17,18].

These observations indicate that the CSF1R-dependent postnatal expansion of tissue-resident macrophages is crucial to development. We previously inferred the specific

expression profiles of rat microglia and brain macrophages, liver macrophages (Kupffer cells) and marginal zone macrophages of the spleen by comparing tissue expression profiles from wild-type and macrophage-deficient *Csf1rko* rats [15,17,19]. In the current study we extend transcriptomic profiling of juvenile wild-type and *Csf1rko* rats to all major organs. The analysis enables the extraction of a generic macrophage transcriptomic signature including C1Q and other collectins as well as tissue-specific CSF1R-dependent signatures that are restored by BMT. We use these data to identify tissue-specific impacts of macrophage deficiency and evidence of functional redundancy and to develop a holistic model of the molecular basis of failure-to-thrive in macrophage-deficient animals. These studies have implications for understanding and treating human CSF1R deficiency and also highlight the potential of CSF1 as a treatment to promote innate immune development in premature or low birth weight infants.

## Results

### A transcriptomic atlas of juvenile wild-type and *Csf1rko* Dark Agouti rats reveals a core CSF1R-dependent signature

To provide a framework for understanding the effect of macrophage deficiency in *Csf1rko* rats and to establish a baseline for the phenotypic rescue by BMT, we isolated mRNA from lung, heart, multiple regions of intestine, kidney, lymph nodes, thymus, adipose, skeletal muscle and diaphragm, testis, ovary and adrenals of 3 week old male and female WT and *Csf1rko* rats, generated deep total RNA-seq data and quantified gene expression. Our analysis includes re-mapping of the previous datasets generated from pituitary and multiple brain regions [19] and from liver [17]. This dataset can be viewed interactively at https://github.com/ dylancartercusack/RNA-Seq-Web-App.

Fig 1A shows a network graph for the sample-to-sample relationships of non-lymphoid tissues. Highlighting the quality of the replicates, samples from each tissue cluster closely with each other. Consistent with previous analysis [17] the liver of juvenile *Csf1rko* rats is clearly distinguished from WT whereas there is little distinction between brain regions based upon genotype [19]. Amongst all of the other tissues profiled, only in pituitary was the overall transcriptomic profiles sufficiently divergent to separate *Csf1rko* and wild-type. We then performed a network analysis of the non-lymphoid tissues using Graphia [20]. By contrast to the widely-used weighted gene correlation analysis (WGCNA) [21], Graphia generates a true correlation graph from which coexpression modules are defined using the Markov clustering (MCL) algorithm. Outcomes are similar to WGCNA, but the co-expression modules are biologically enriched compared to those defined by WGCNA [20].

Fig 1B shows the network graph for gene-to-gene relationships across the non-lymphoid tissue dataset. Co-expression modules generated with an r value of ≥ 0.8 and MCL inflation value (determining granularity of the clusters) of 1.3 are shown in S1 Table with average expression profiles for each cluster containing >5 genes. The analysis complements and greatly extends a previous rat RNA-seq body map in the Fischer 344 background that included 2 week old juvenile animals [22] which were included in a larger meta-analysis [23]. As expected, we identified co-expressed clusters of genes that are either tissue-specific (examples shown in Fig 1C) or process/pathway specific and in each case we are able to determine that the *Csf1rko* has no significant effect on the average expression of genes in each cluster. The obvious exception was Cluster 21, which contains *Csf1r* and the set of genes that is CSF1R-dependent in every non-lymphoid organ. The average expression of this cluster and the gene list is shown in Fig 2A. By contrast to the mouse, where MHCII genes define subsets of tissue macrophages [24], and in common with humans, rat macrophages aside from microglia express high levels of MHCII genes (e.g., *RT1-DMa, Cd74* and the transcription factor *Ciita*).

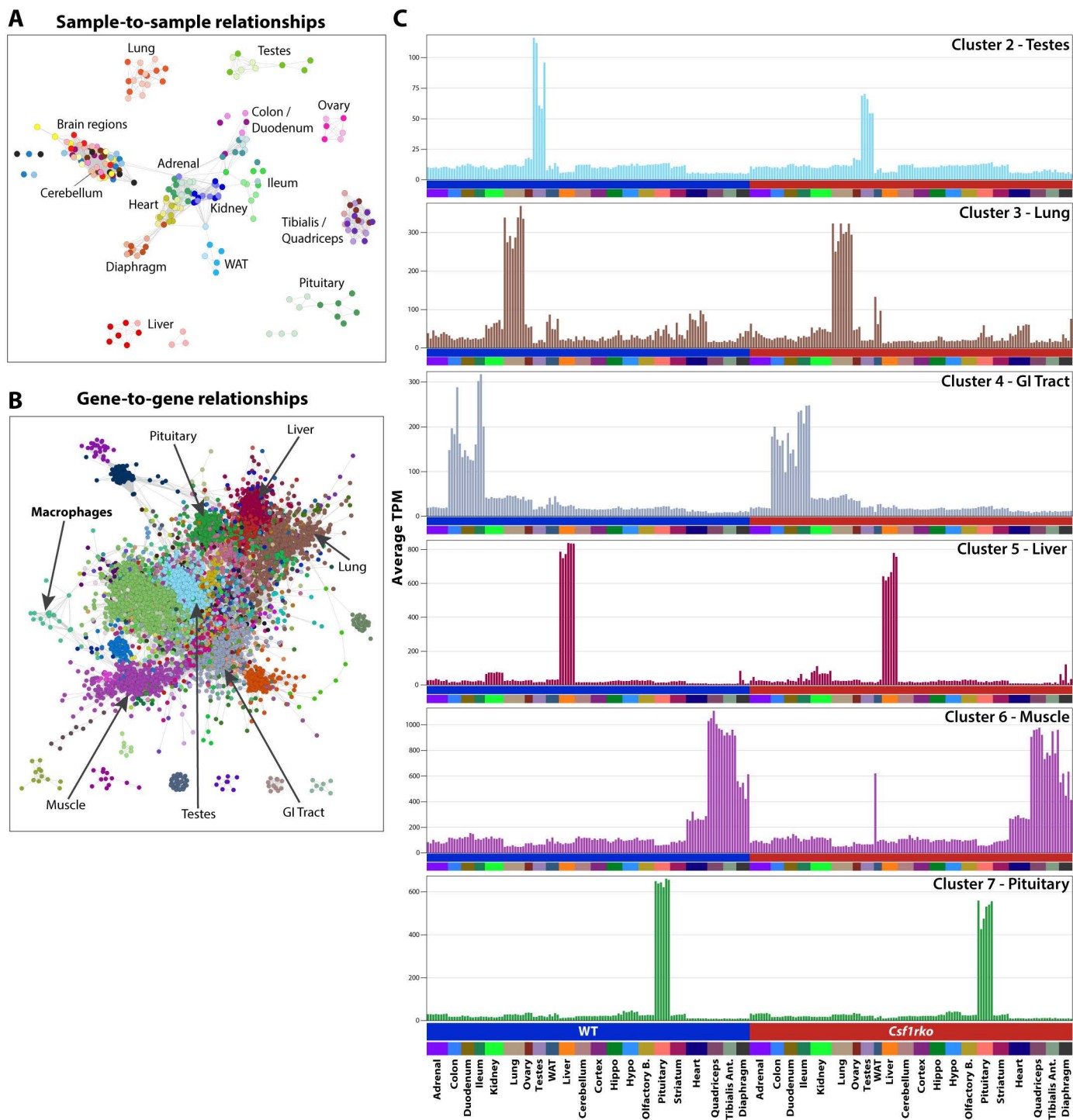

**Fig 1. The effect of the *Csf1rko* on tissue-specific gene expression in juvenile rats.** Networks derived from tissue RNA−Seq data from 3-wk WT and *Csf1rko* rats. Visualisations were created in BioLayout. **(A)** Sample-to-sample network graph created in Biolayout using a Pearson correlation coefficient threshold of r > 0.8. Individual samples (nodes) are connected by edges (lines) that reflect the corresponding r value. Nodes are coloured by organ, with a darker shade for WT and a lighter shade for *Csf1rko*. **(B)** Gene-to-gene network graph was created in Biolayout for all non-lymphoid tissues; correlation coefficient threshold of r > 0.8 and an MCL inflation value of 1.3. Nodes represent genes and nodes with similar expression patterns (clusters) are shown in the same colour. **(C)** Example cluster expression plots showing the average expression of genes contained in each cluster. Each column shows the tissue indicated on the X axis for a different animal. The Y axis shows the average expression of genes in the sample in gene-level transcripts per million (TPM).

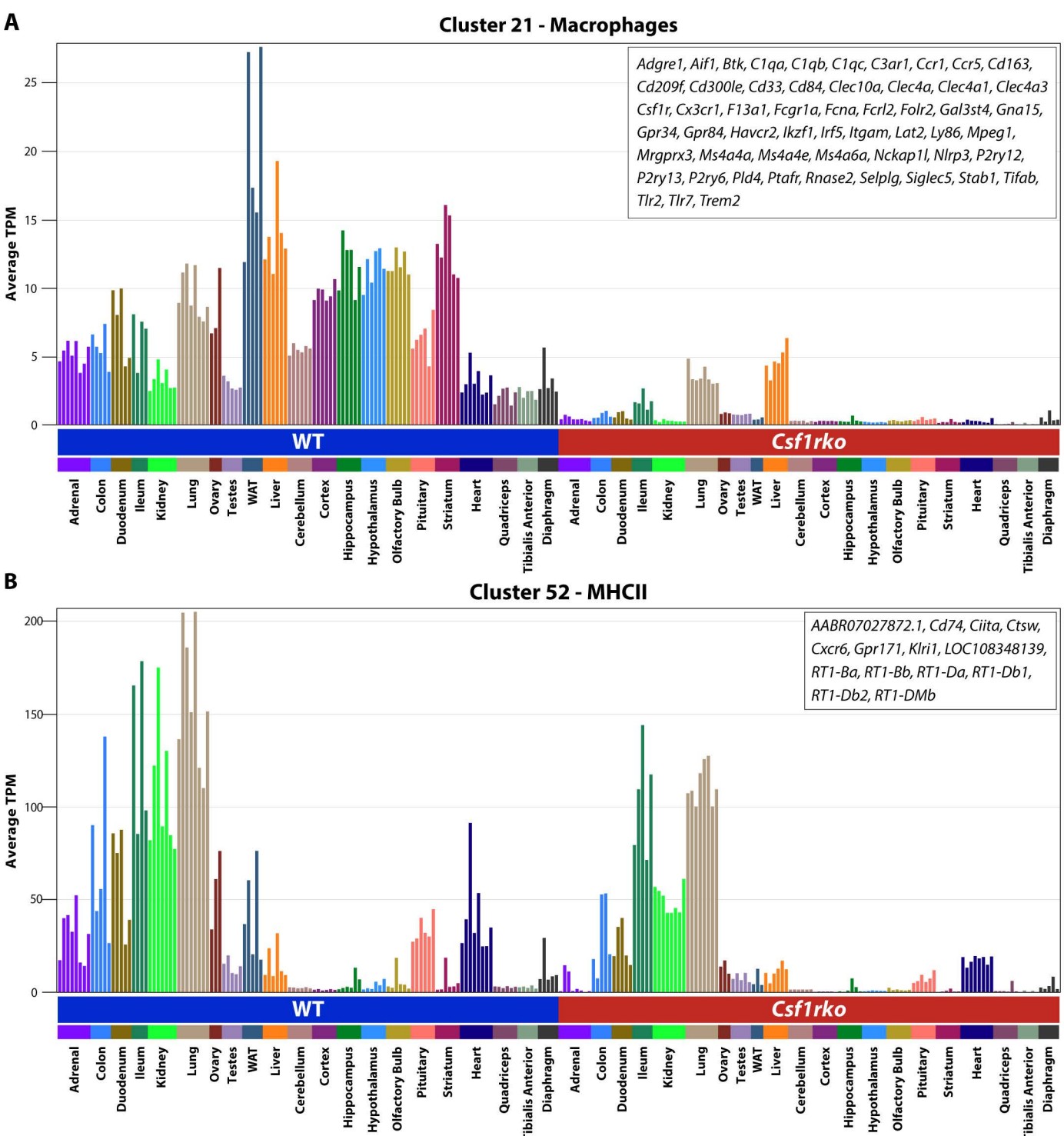

**Fig 2. Identification of *Csf1r* co-expressions cluster.** S1 Table identifies clusters of genes that share co-expression across non-lymphoid tissues of 3 week old WT and *Csf1rko* based upon gene-to-gene pair-wise correlation analysis in Graphia. The analysis revealed two clusters that distinguish WT and *Csf1rko* tissues. The Figure shows the average expression of **(A)** cluster 21 – Macrophages (55 genes) and **(B)** cluster 52 – MHCII related genes (14 genes). Each column shows the tissue indicated on the X axis for a different animal. The Y axis shows the average expression of genes in the sample in gene-level transcripts per million (TPM). The list of genes contained in each cluster is shown in the top right of each plot.

These genes form a small cluster, Cluster 52. The average profile shows that the genes in Cluster 52 are CSF1R-dependent to varying degrees in every non-lymphoid tissue (Fig 2B).

## Analysis of individual tissues reveals tissue specific macrophage signatures and identifies redundant and non-redundant functions during development

The core cluster of universal CSF1R-dependent genes (Cluster 21) excludes genes affected by the *Csf1rko* in only one or limited tissues. For example, Kupffer cell (KC) markers *Clec4f, Vsig4* and *Cd5l* [25] are within the liver-specific cluster. They were reduced by 60-70% in the juvenile *Csf1rko* liver, consistent with persistence of a CSF1R-independent population that nevertheless retains KC identity [17]. Network analysis allows tissue RNA-seq data to be deconvoluted to infer the relative abundance of specific cell types based upon cell type-specific markers [24]. Analysis of the profiles of six brain regions comparing juvenile wild-type and *Csf1rko* revealed a microglia-specific expression signature and the lack of impact of microglia deficiency on region-specific gene expression [19]. Such an analysis is expedited by having multiple replicates, which vary by chance in their relative cellular composition, as well as different conditions that alter abundance of specific cell types. To further dissect the effect of macrophage deficiency on postnatal organ development, we performed network analysis on the individual tissue RNA-seq data sets. In the sections below cluster numbers refer to the tissue specific network analyses.

## Lung

*Csf1rko* rats require euthanasia by around 8 weeks of age, often due to progressive breathing difficulties. Shibata *et al.* [26] described progressive emphysematous changes in the lung in *Csf1*<sup>op/op</sup> mice and focussed on delayed development of alveolar macrophages (AM) as the underlying mechanism. Examination of *Csf1rko* rat lungs at 3 weeks of age revealed destruction of alveolar walls, reduced alveolar numbers and expansion of the terminal airways in *Csf1rko* lung compared to control (S1 Fig), similar to the reported phenotype in older *Csf1*<sup>op/op</sup> mice. The mouse lung contains multiple mononuclear phagocyte populations defined as recruited monocytes, conventional dendritic cells, bronchoalveolar macrophages and interstitial macrophages based upon surface markers [27]. There have been many studies aimed at defining subpopulations of lung interstitial macrophages in mice, but the analyses are confounded by selective recovery and activation of the cells during tissue disaggregation [28]. In mice, the major growth factor required for maintenance of bronchoalveolar macrophages is CSF2 (granulocyte-macrophage colony-stimulating factor). In our dataset, *Csf2* mRNA expression was almost completely lung-specific, but both *Csf1* and *Il34* mRNA were also highly-expressed in the lung, perhaps explaining a more penetrant lung phenotype in the *Csf1rko* than in CSF1-deficient (*Csf1*<sup>tl/tl</sup>) rats [29]. Cluster analysis revealed only minor impacts of the *Csf1rko* on lung development at 3 weeks of age, despite the apparent pathology (S2 Table). Abundantly expressed markers of alveolar type 1 (AT1) cells (e.g., *Ager, Pdpn1*) and type 2 (AT2) cells (*Sftpa1, Sftpc, Lyz2*) [30] were not affected by *Csf1rko*. Consistent with previous analysis of Ki67 staining [17], **Cluster 5,** which contained genes reduced in *Csf1rko* lung included numerous cell-cycle related genes, whereas there was a reciprocal small increase in ribosomal/protein synthesis genes (**Cluster 7**). The set of 73 genes correlated with *Csf1r* (**Cluster 11**) identified macrophage-associated genes that were depleted to different extents in the *Csf1rko* lung. **Cluster 28** adds a further 21 genes with a similar pattern. Highly depleted genes including *Adgre1, C1qa/b/c, Cd4, Cd163, Cd300le, Clec4a, Clec10a, Cx3cr1, Folr2, Hpgds, Ltc4s, Pld4* and *Sh2d6* were reduced by 80% or more. Each of these genes is expressed

at very low levels in isolated rat alveolar macrophages [31] supporting the conclusion that they represent an interstitial macrophage signature. Fig 3A shows examples of the downregulation of these selected genes in *Csf1rko* lung.

The effect of the *Csf1rko* on detection of other myeloid-expressed genes in lung was variable. A cluster of highly-expressed neutrophil-specific genes (**Cluster 12**), including *Camp, Mmp8, Mmp9, Ngp*, *S100A8* and *S100A9,* showed increased expression in the *Csf1rko* animals. Gene set enrichment analysis (GSEA) confirmed this elevation in neutrophil related genes (Fig 3C). This may be associated with the granulocytosis discussed further below but may also relate to increased expression of proinflammatory cytokines *Il1b* and *Cxcl1,* and the emphysema-like pathology that develops in these animals. The dendritic cell growth factor gene *Flt3lg* is highly-expressed in the juvenile rat lung and markers associated with cDC1 (type 1 classical dendritic cells), including *Batf3, Ccr7, Clec9a, Flt3, Itgae* and *Xcr1* [24] were not significantly affected by the *Csf1rko* (Fig 3B). Similarly, rat alveolar macrophage markers defined previously [31], including *Cd2, Clec4d, Clec7a, Csf2ra/b, Fabp4/5, Mcoln3, Mrc1, Pparg* and *Siglec5* were unaffected (Fig 3B), consistent with lack of effect of the *Csf1rko* on this population [17]. Other myeloid-associated genes, including *Fcgr1a, Irf8, Itgam, Siglec1, Spi1* and class II MHC (*RT1-DMb* etc), were reduced to varying extents. Many of these genes are likely shared by multiple macrophage, DC and granulocyte populations retained or expanded in mutant lung which may obscure the effect of the loss of CSF1R-dependent macrophages on total tissue gene expression.

To confirm the existence of a CSF1R-dependent interstitial macrophage population, we examined flow cytometry profiles of disaggregated lungs from wild-type and *Csf1ko* rats. Fig 3D identifies a major population of CSF1R-dependent cells, distinct from AM, defined by relative expression of CD11b/c and CD172a. There was also a substantial increase in CD11b/c-Hi granulocytes, consistent with the RNA-seq data. The infiltration of these cells, which also express CD172a, further indicates why genes encoding these markers (*Itgam, Itgax, Sirpa*) and other myeloid-specific genes are not reduced in the bulk RNA-seq data.

One constraint on study of the rat as a model is the limited availability of well-defined antibodies suitable for immunohistochemistry [32]. One widely studied marker is the haptoglobin receptor, CD163. CD163 was detected on a subset of macrophages in the rat embryo, notably associated with erythroblastic islands in the liver, and expression was abolished in *Csf1rko* embryos [16]. *Cd163* mRNA was detected in WT and absent in *Csf1rko* lung. Consistent with the original study localising this marker in rat lung [33], CD163 was detected exclusively on macrophage-like cells (red arrows) within connective tissue surrounding the major airways (red asterisks) and blood vessels and was undetectable in *Csf1rko* lung (Fig 3E).

## Lymphoid tissues

Lymphoid tissues were omitted from the global tissue clustering analysis because their unique expression profiles overpowered the differences observed in other organs. Although CSF1 has been implicated as a local trophic factor for macrophages in the splenic red pulp in mice [34], previous analysis of the spleen of adult *Csf1rko* rats revealed only partial reduction in this population and associated genes involved in red cell clearance and iron metabolism [15]. By contrast, localisation of the marginal zone metallophilic macrophage marker, CD169 (*Siglec1*), alongside microarray profiling, indicated the selective loss of this population. The expression data also inferred depletion of the separate marginal zone macrophage population defined by the markers *Cd209* and *Cd209b* [15] encoding lectin-like receptors DC-SIGN and SIGNR1 respectively. Like the structurally-related MRC1 (CD206), CD209 receptors bind mannose-containing microbial structures [35] and like CD206, CD209B is expressed in the liver by sinusoidal endothelial cells (LSEC) as well as macrophages [36].

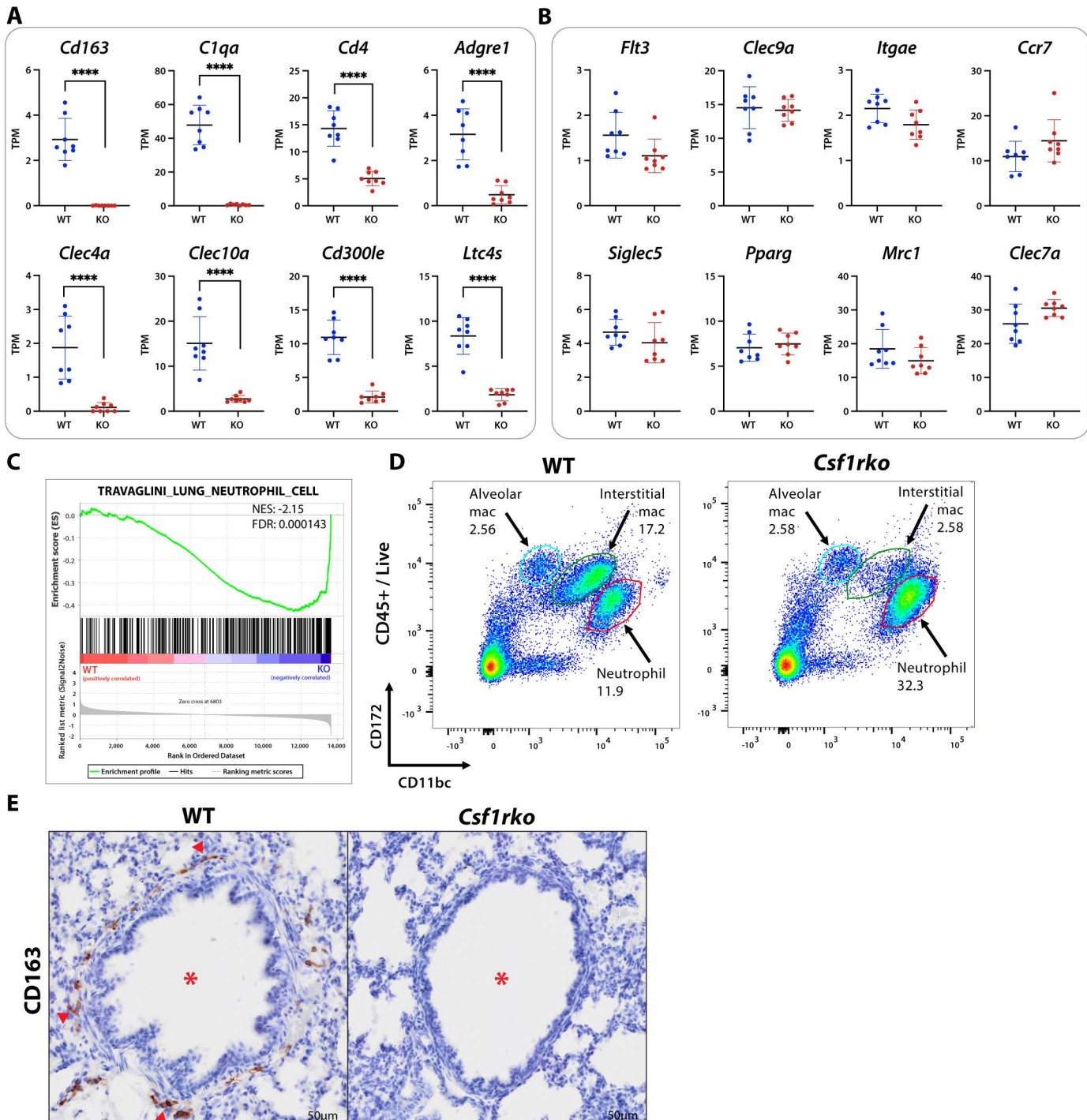

**Fig 3. Selective loss of interstitial macrophages in the lung of juvenile *Csf1rko* rats.** (A–B) Expression in transcripts per million (TPM) for selected genes from RNA-Seq data of 3wk WT and *Csf1rko* lungs (n = 8 WT; 8 *Csf1rko*). Graphs show the mean ± SD. (C) Gene set enrichment plot created using the C8 (cell type signature) gene set collection from the Molecular Signatures Database (MSigDB). NES – Normalised Enrichment Score; FDR – False Discovery Rate. A negative enrichment score indicates an enrichment in the *Csf1rko* animals. (D) Representative flow cytometry plots from disaggregated whole lung showing differential expression of CD172 and CD11bc. (E) Representative images showing immunohistochemical localization of CD163 (brown) in 3 week WT and *Csf1rko* lungs. * indicates primary airway bronchiole. ▲ indicates examples of CD163+ macrophages. Original magnification: 40X. Scale bar: 50μm. *, P < 0.05; **, P < 0.01; ***, P < 0.001; ****, P < 0.0001.

Immunohistochemical localisation using an antibody against CD209B confirmed the expression of this protein in LSEC in rat liver (S2A Fig). In the rat spleen, as in mouse [37], CD209B was localised to a stellate macrophage-like population in the marginal zone. Consistent with previous expression profiling of the spleen indicating complete loss of *Cd209b* mRNA [15], the protein was undetectable in the *Csf1rko* spleen (Fig 4A).

To extend the analysis of spleen, we compared profiles of lymph nodes and thymus from juvenile wildtype and *Csf1rko* rats and conducted network analysis on them together. Co-expressed gene clusters across these tissues are shown in S3 Table. Two of the largest clusters were tissue-specific, **Cluster 1** was enriched in thymus and contains T cell-specific genes (e.g., *Cd3*) and genes involved in proliferation and differentiation and **Cluster 3** contains B cell-specific genes (e.g., *Cd19*). Despite the relative depletion of circulating B cells, and early thymic involution in *Csf1rko* rats, neither cluster was affected by the *Csf1rko*. **Clusters 12** and **98** contained *Csf1r* and 117 other genes that were detected at higher levels in LN than thymus and which were CSF1R-dependent. Fig 4B highlights the regulation of numerous endocytic pattern recognition receptors of multiple classes. The products of these genes contribute the role of LN macrophages in sentinel function and antigen capture which is considered in more detail in the discussion. Many of the genes were only partly CSF1R-dependent, presumably reflecting the residual mononuclear phagocyte populations detected by IBA1 (*Aif1* gene) staining (S2B Fig), especially within T cell areas [18]. Amongst the genes detected in LN, as in the spleen, *Cd209b* was highly-expressed and undetectable in *Csf1rko*. In mouse LN, CD209B was detected on a population of medullary sinus macrophages [37]. Similarly, we detected abundant CD209B immunoreactivity in the medullary and subcapsular sinuses that was absent in the *Csf1rko* LN (Fig 4A).

The thymus contains a population of macrophages involved in phagocytosis of apoptotic thymocytes. In mouse, Zhou *et al.* [38] defined two thymic macrophage populations based upon expression of *Timd4* and *Cx3cr1*. Analysis of total RNA-seq data in the thymus is complicated by the early onset of involution in the *Csf1rko*. In some animals the thymus was already almost completely absent at 3 weeks of age. Fig 4C highlights up and down-regulated genes comparing WT and *Csf1rko*. Aside from *Cd209*, *Cd209b* and *Cd209f* which were absolutely CSF1R-dependent, as noted in splenic red pulp [15] and LN, other macrophage-associated genes (e.g., *Aif1,C1qa/b/c, Folr2, Mrc1*) were reduced to varying degrees suggesting the presence of a CSF1R-independent macrophage population, as reflected by IBA1 staining (S2B Fig). By contrast to the profile reported for isolated mouse thymic macrophages [38], *Cx3cr1*, *Timd4* and the transcription factor *Spic* were detected in rat thymus but were not CSF1R-dependent (Fig 4C).

Considering the thymus in isolation, there is a subset of macrophage-expressed genes, including *Cx3r1, Sirpa* and transcription factors *Spi1* and *Mafb*, that was increased in *Csf1rko* thymus, likely indicating an expansion of active CSF1R-independent phagocytes involved in involution. The set of up-regulated genes in the *Csf1rko* thymus (**Cluster 6**) includes *Foxn1*, the regulator of thymic epithelial differentiation [39] and cytokeratins (e.g., *Krt5*) perhaps indicating a feedback mechanism to mitigate involution. This may explain the 2-fold increased detection of *Ciita, Cd74* and class II MHC genes which are expressed by thymic epithelium. Given the absolute loss of *Cd209b* mRNA we stained thymus (Fig 4A) and identified a novel population of CD209B+ macrophages mainly located in the cortex and corticomedullary junction (but not present in the medulla) which were absent in the *Csf1rko*.

## Gastrointestinal tract

The GI tract contains one of the most abundant macrophage populations in the body. As in other tissues, there are specific niches within the tissue, notably the lamina propria, submucosa and muscularis. GI tract macrophages in mouse are believed to turn over relatively

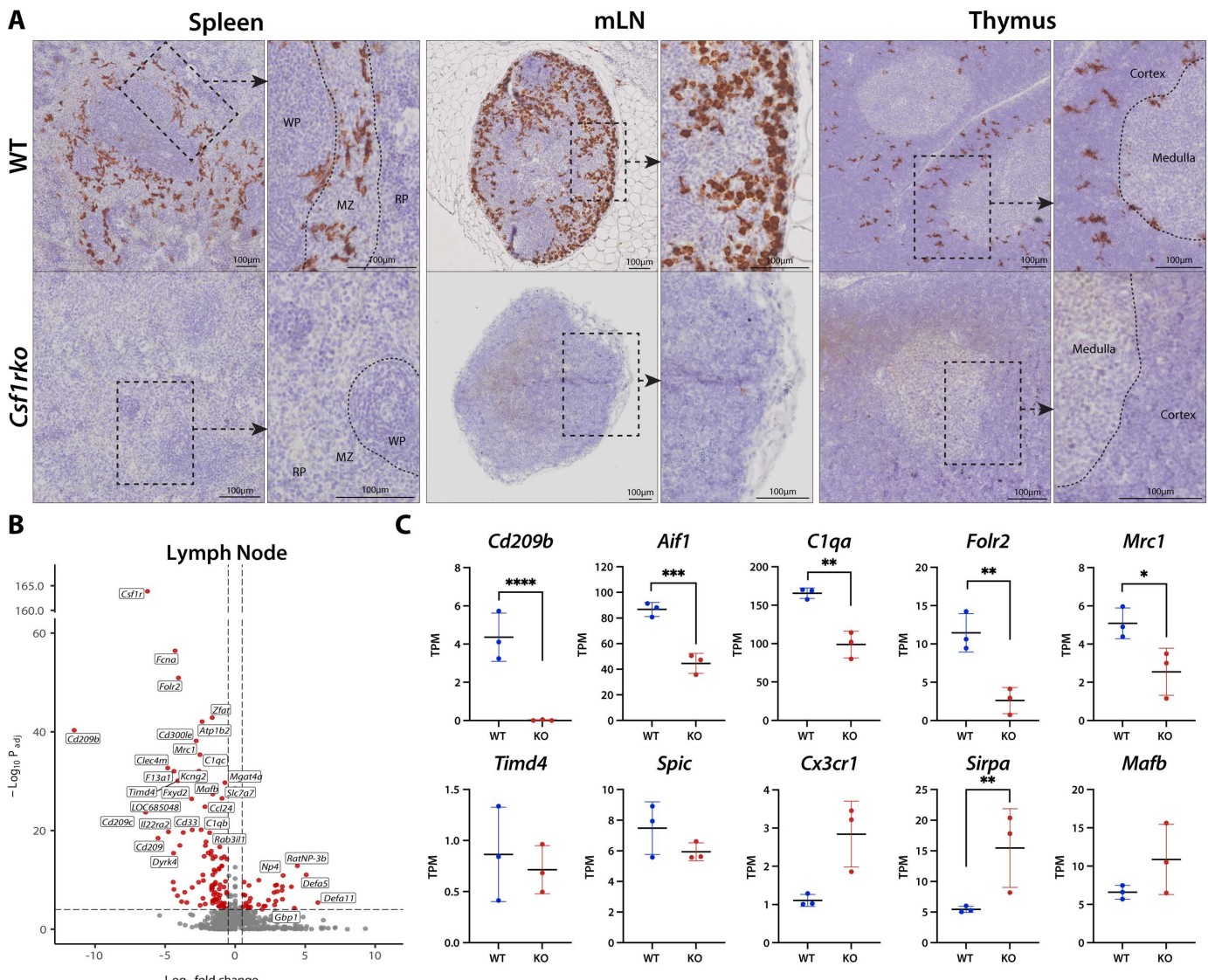

**Fig 4. Selective loss of macrophage subpopulations in lymphoid tissue of *Csf1rko* rats.** (A) Representative images showing immunohistochemical localization of CD209B (brown) in 3 week WT and *Csf1rko* spleen, mesenteric lymph nodes (mLN) and thymus. Original magnification: 40X. Scale bar for main images: 100μm. Dashed boxes indicate the zoomed in region shown beside the main image. Annotations – WP: white pulp, MZ: marginal zone, RP: red pulp. (B) Volcano plot showing differential expression (DE) between 3 week WT and *Csf1rko* lymph nodes (n = 12 WT; 11 *Csf1rko*). DE was calculated using DESeq2, and log fold changes (logFC) were shrunk using lfcShrink. Positive logFC indicates an increase in expression in *Csf1rko* animals, while negative logFC indicates a decrease in expression in *Csf1rko* animals. (C) Expression profiles in gene-level transcripts per million (TPM) for selected genes from RNA-Seq data of 3wk WT and *Csf1rko* thymus (n = 3 WT; 3 *Csf1rko*). Graphs show the mean ± SD. *, P < 0.05; **, P < 0.01; ***, P < 0.001; ****, P < 0.0001.

rapidly and to be replaced by monocytes [40], but a subset may be relatively long-lived and capable of self-renewal [41,42]. Macrophage depletion studies have implicated resident macrophages in control of enteric neurons, vascular homeostasis and intestinal stem cell differentiation [43–47]. Many of the functions are influenced by the microbiota and might differ between the relatively sterile duodenum and the lower GI tract. In the mouse, lamina propria macrophages defined by expression of *Cx3cr1* have been reported to increase from small intestine to colon [46]. The combined cluster analysis of the duodenum, ileum and colon is

provided in Tab A in S4 Table. Some of the clusters are relatively enriched in individual samples, likely reflecting differential sampling/contribution of substructures including isolated lymphoid follicles and follicle-associated epithelium, neuroendocrine cells, crypts/Paneth cells and pancreatic enzymes in the duodenum.

As in the lung, IBA1 staining indicated that the *Csf1rko* ileum contained some residual lamina propria mononuclear phagocytes [17]. The expression profiles suggest these cells are monocytes, since markers such as *Cd14, Cxcr4,* and *Nod2* were relatively unaffected in the *Csf1rko*. By contrast to mouse, where it is considered a functional marker of lamina propria macrophages [48], *Cx3cr1* was not highly-expressed in the rat intestine. Amongst other intestinal macrophage markers identified in mouse [41,42], *Timd4* was not reproducibly detectable and *Lyve1* was abundant but unaffected by the *Csf1rko*. **Cluster 23** identified the relatively small set of 44 genes with similar profiles to *Csf1r* across the three gut regions (Fig 5A). Within this cluster, the genes follow slightly different patterns of expression. The most globally CSF1R-dependent genes including *Adgre1, C1qa/b/c, Cd68, Cd163, Clec10a, Fcgr1a* and *Msr1* were detected at similar levels in all three regions and were completely lost in the *Csf1rko* (Fig 5B). However, several genes (e.g., *Aif1, Cd68, class II MHC, Tyrobp*) were depleted to a greater extent in colon and duodenum compared to ileum (Fig 5C). This pattern suggests that ileum, like lung, contains a CSF1R-independent mononuclear phagocyte population with a distinct expression profile. This may include classical dendritic cells. *Flt3lg* mRNA and markers of type 1 classical dendritic cells (cDC1) identified in the mouse ([24]; *Clec9a, Flt3, Xcr1*) were increased in the ileum in the *Csf1rko* (Fig 5D). Tabs B-D in S4 Table contain the profiles of myeloid-specific genes and other genes of interest from each region. *Siglec1*, which defines a crypt-associated macrophage population in mice [43] was also detected and CSF1R-dependent. Across all three regions there were few genes that would constitute a CSF1R-dependent gut-specific macrophage signature. One candidate is *Adgrg5*, encoding an adhesion G protein-coupled receptor (GPR114) that was relatively enriched in intestine and was CSF1R-dependent. Given the rapid turnover of intestinal macrophages from monocytes [41,42], the CSF1R-dependent expression of the chemokine receptor, *Ccr1*, in all gut regions is likely to reflect a function in recruitment.

Intestinal macrophages are proposed as a constitutive source of IL10 [46] although there is contrary evidence from conditional KO in mice [49]. *Il10* mRNA was barely detected in rat intestine and not significantly altered in *Csf1rko*. A recent study in mice claimed that C1Q-expressing macrophages were located specifically in the intestinal submucosal and myenteric plexuses and associated with enteric neurons [50]. This is not consistent with the high level expression of C1Q subunits and the absolute CSF1R-dependence we observed in the rat.

Amongst the functions attributed to macrophages, we found no evidence to support selective effects of the *Csf1rko* on epithelial proliferation (**Cluster 2**) or region-specific differentiation (**Clusters 1, 5, 11**). By contrast to previous analysis of *Csf1rko* mice [51] the relative abundance of major cell types defined by marker genes (e.g., *Chga*, neuroendocrine cells; *Muc2/3/13*, goblet cells; *Lgr5* stem cells) was not significantly altered in any of the three regions. We were interested specifically in the proposed role of macrophages in the pruning of peripheral nerves during intestinal nervous system development [44,45,50] in view of evidence that similar functions attributed to microglia in the brain appear to be redundant [19]. **Cluster 8** contained multiple markers of enteric neurons, including *Ngfr*, neurofilament heavy and light chains, and the intermediate filament protein peripherin (*Prph*) that were significantly increased in both ileum and colon. To assess the relationship to enteric neuronal development, sections of ileum and colon were stained for peripherin, which is expressed specifically within nerves of the peripheral nervous system. Fig 5E shows an increase in the size of neuronal tracts running throughout the myenteric and submucosal plexus in both tissues

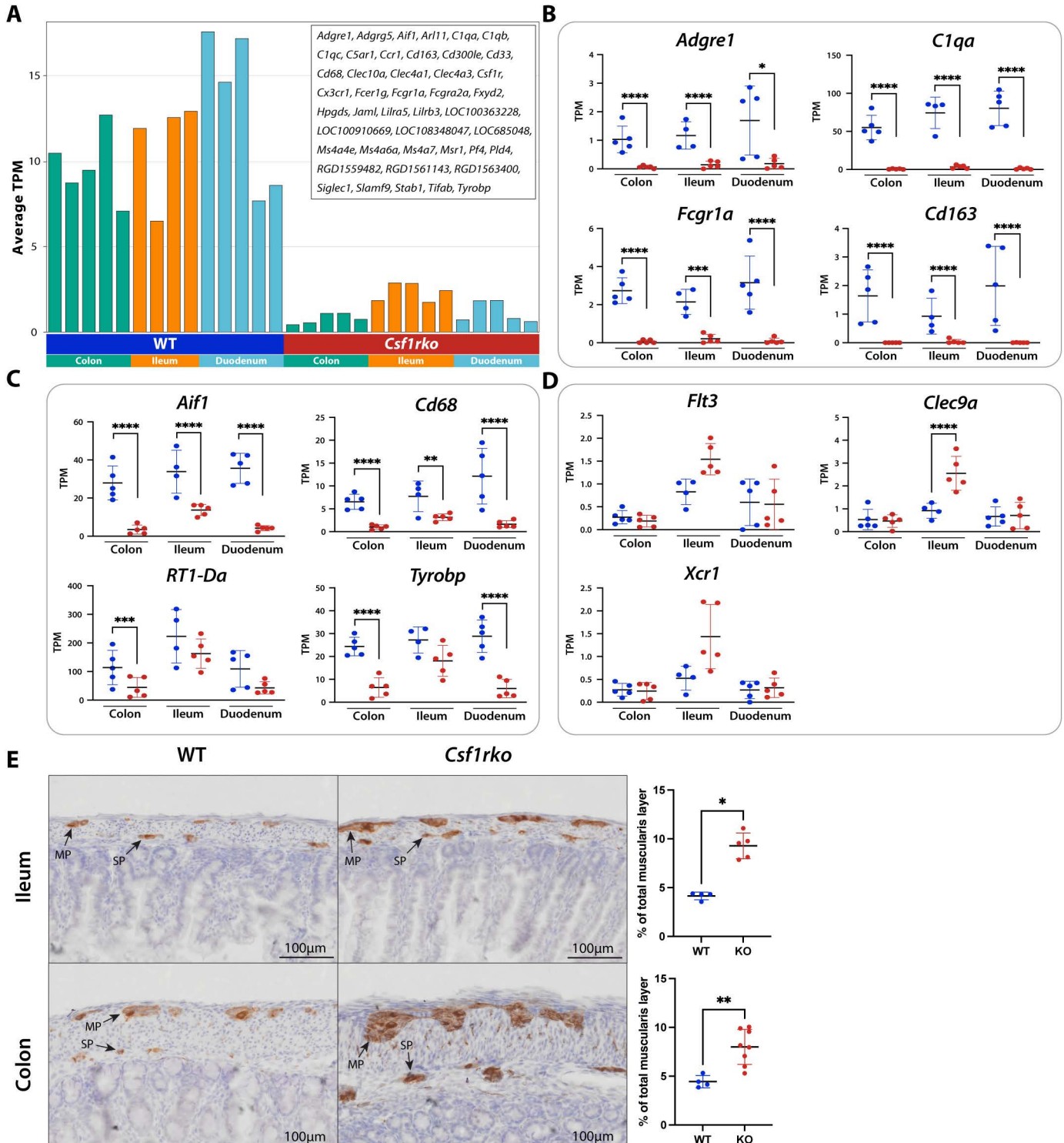

**Fig 5. The effect of the *Csf1rko* on gene expression and neuronal development in the gut.** Gene expression profiles of colon, ileum and duodenum from WT and *Csf1rko* rats were clustered using Graphia (S4 Table). **(A)** The profile of Cluster 23 containing 44 genes. Y-axis shows the average expression of transcripts in the cluster (TPM); individual columns are biological replicates of the WT and *Csf1rko* tissues as indicated. **(B–D)** Expression profiles in transcripts per million (TPM) for selected genes from RNA-Seq data of 3wk WT and *Csf1rko* colon, ileum and duodenum (n = 4-5/group). Graphs show the mean ± SD. **(B)** *Csf1r*-dependent genes common to all three tissues. **(C)** *Csf1r*-dependent genes depleted to a greater extent in colon and duodenum compared to ileum. **(D)** DC-associated genes increased in the *Csf1rko* ileum. **(E)** Representative images showing immunohistochemical localization of peripherin (brown), a marker of enteric

neurons, and quantification of this staining in 3 week WT and *Csf1rko* ileum and colon. MP: myenteric plexus, SP: submucosal plexus. Original magnification: 40X. Scale bar: 100μm. The relative positive stain area for peripherin as a proportion of the muscularis and submucosal layer was determined by quantitative image analysis of an entire Swiss roll (n = 4 WT; 5 *Csf1rko* biological replicates). Graphs show the mean ± SD. *, P < 0.05; **, P < 0.01; ***, P < 0.001; ****, P < 0.0001.

of the *Csf1rko*. This supports the suggestion that macrophages are involved in the refinement of intestinal neurons during early postnatal development [45], and their absence in *Csf1rko* animals allows the neurons to grow unchecked.

## Muscle

**Cluster 4** in the intestine analysis contained the expression signature of smooth muscle (e.g., *Lmod1*, *Myl9*, *Actg2*, *Dmpk*, *Acta2*, *Flna*) (Tab A in S4 Table). The contribution of this cluster varied between samples irrespective of genotype, suggesting that the lack of muscularis macrophages does not prevent normal smooth muscle development. In mouse skeletal muscle, muscle fibre number is determined at birth and postnatal growth occurs mainly by hypertrophy but in rats both hyperplasia and hypertrophy contribute [52]. Reflecting the difference in somatic growth, the skeletal muscles of *Csf1rko* pups are smaller than WT, and the average fibre diameter is 30-40% reduced [17]. We compared gene expression in diaphragm, tibialis anterior (TA), quadriceps and heart. Co-expression clusters are shown in Tab A in S5 Table. Three large clusters are associated with the *Csf1r* genotype and are consistent with development delay in the *Csf1rko*. These are highlighted in **Fig 6A-C.** The largest, **Cluster 1**, grouped 1572 genes that are enriched in all three skeletal muscles relative to heart and expressed more highly in the *Csf1rko* (Fig 6A). This cluster includes the major fetal growth factor, *Igf2.* Gene ontology (GO) term annotation revealed strong enrichment for protein synthesis in this cluster. We interpret this to indicate ongoing hypertrophy in the *Csf1rko* that has declined in the WT. **Cluster 4** had a similar profile but was more skeletal muscle-specific, containing multiple myogenic transcription factors, including *Myod1*, a signature of satellite cells (Fig 6B) [53]. This cluster shared increased detection in the *Csf1rko* indicating ongoing generation of myonuclei. Finally, **Cluster 5**, also skeletal muscle-specific, was reduced in the *Csf1rko*, and GO term annotation showed enrichment of genes involved in mature muscle function including many of the enzymes of glycolysis (Fig 6C). The entire GO term analysis for these 3 clusters can be found in Tabs B-D in S5 Table. By contrast, genes in **Cluster 2,** strongly enriched in the heart, and genes encoding enzymes of oxidative phosphorylation (**Cluster 14**) were unaffected in *Csf1rko* animals. **Cluster 19** contained 51 macrophage-specific genes that, like *Csf1r*, were almost undetectable in *Csf1rko* muscle. The larger CSF1R-dependent **Cluster 9** contained additional myeloid genes (e.g., *Laptm5*, *Spi1*, *Spn*, *Tyrobp*) as well as class II MHC genes and NK cell receptors that were expressed more highly in heart relative to skeletal muscle.

Postnatal innervation in rat muscle proceeds through a stage of polyneuronal innervation. By the end of the second week of life, this resolves to a single motor neuron axon for each muscle fibre [54]. Peripheral neuronal markers that were increased in the intestine (e.g., *Prph*) and acetyl cholinesterase (*Ache*), a marker of neuromuscular junctions, were not affected in the *Csf1rko*. The absence of microglia in a mouse hypomorphic *Csf1r* mutation was associated with hypermyelination of axons in the central nervous system [55]. In the periphery, axon myelination is mediated by Schwann cells. Schwann cell maturation and peripheral myelination in rodents occurs in the immediate postnatal period and is largely complete by the time of weaning [56–58]. **Cluster 8** contains transcripts encoding myelin-associated genes, key transcriptional regulators (e.g., *Sox 10*) and markers of immature Schwann cells [56–58]; Fig 6D shows the cluster co-expression plot along with key examples of these markers which are enriched 2-3 fold in the *Csf1rko* muscle. The literature on macrophage-Schwann cell

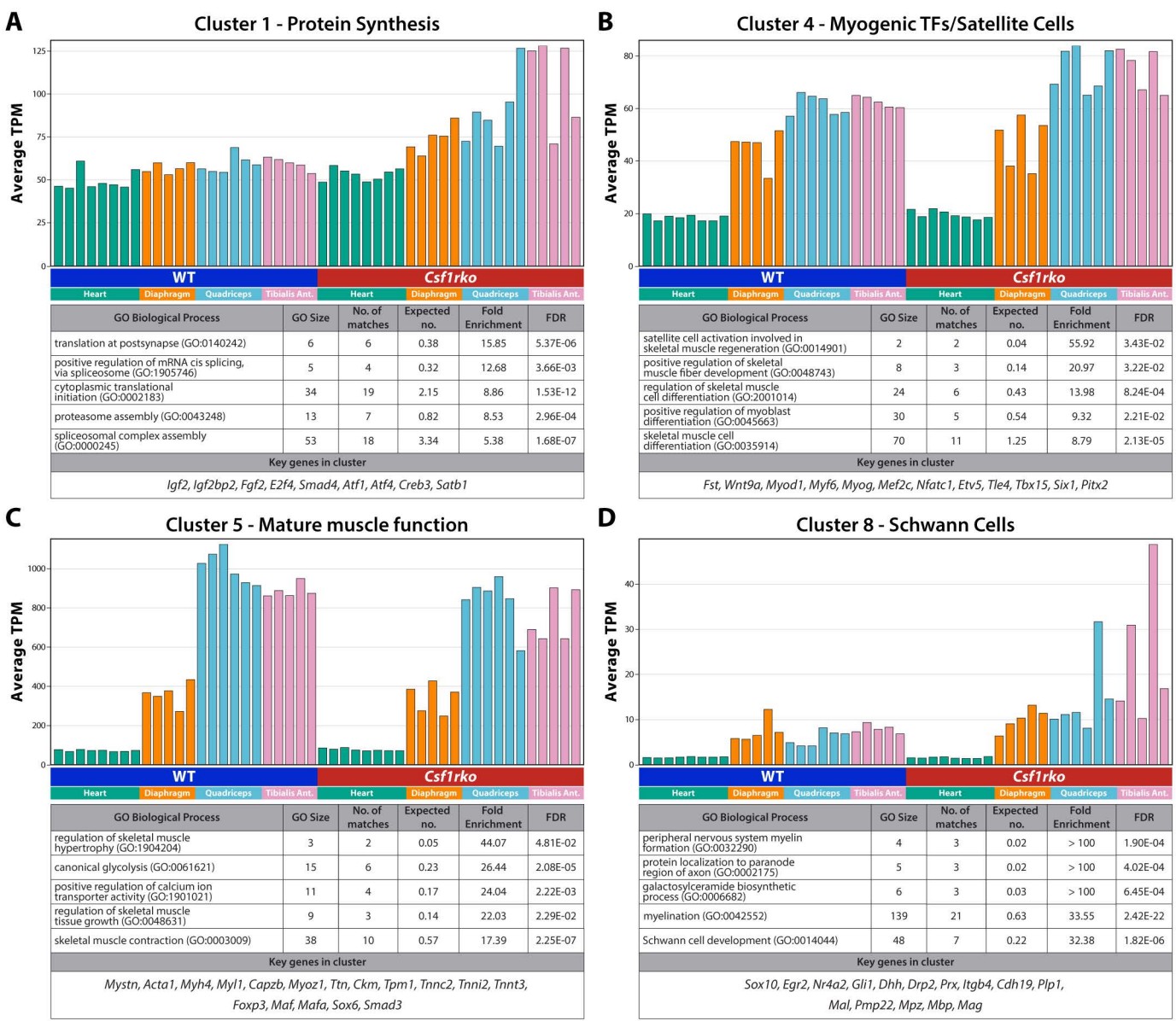

**Fig 6. The effect of the *Csf1rko* on muscle development.** Gene expression profiles of heart, diaphragm, quadriceps and tibialis from WT and *Csf1rko* rats were clustered using Graphia (S5 Table). (A-D) The profiles of Clusters (A) 1(B) 4(C) 5 and (D) 8 from Tab A in S5 Table. Y-axis shows the average expression of transcripts in the cluster (TPM); individual columns are biological replicates of the WT and *Csf1rko* tissues as indicated. Under each expression plot is a table showing (1) key genes contained within each cluster and (2) examples of gene ontology (GO) biological processes that are enriched in the genes contained within each cluster. GO Size: the number of genes contained in each gene ontology; No. of matches: the number of genes from the gene ontology which are found in each respective cluster; expected no. the expected number of matches in a random background sample.

interaction is focussed on remyelination following injury [59]. This pattern suggests a role for macrophages in feedback control of myelination in development.

The diaphragm was of particular interest because of the respiratory phenotype in the *Csf1rko* rats and the loss of circulating IGF1 [17]. *Igf1r* knockout mice had compromised development of the diaphragm, leading to lethal neonatal respiratory distress [60]. As in other skeletal muscle, *Igf1* was unaffected by macrophage deficiency and *Igf2* was highly-expressed

and further increased in the *Csf1rko*. Macroscopic view of the diaphragm in 3 week old rats indicated increased transparency and thinning of the connective tissue of the central tendon, which may indicate underdevelopment or degeneration. There was also inconsistent thinning of the diaphragmatic musculature, which showed a trend in severity with the clinical dyspnoea observed. Consistent with the dysplasia or degeneration of this structure, a unique feature of the diaphragm expression profile was the reduced detection of multiple collagen genes in the *Csf1rko* animals as shown in the gene set enrichment analysis (S3B Fig). Significant haemorrhage was also observed in both the muscular diaphragm and central tendon of 3-week-old *Csf1rko* rats (S3A Fig), which likely contributes to the respiratory phenotype. This is likely related to the increased rigidity of the rib cage in the *Csf1rko* animals, leading to a marked increase in inspiratory pressure, ultimately causing the observed diaphragmatic haemorrhage.

## Kidney and adrenal glands

The initial description of the juvenile *Csf1rko* kidney phenotype noted the relative hypoplasia of the medulla [17]. Consistent with the macroscopic view, cluster analysis revealed increased detection of genes associated with glomeruli and proximal tubules (**Clusters 1** and **11**) and decreased detection of medullary genes (**Cluster 4**) in *Csf1rko* rats (Fig 7A and Tab A in S6 Table). The number of glomeruli was also quantified histologically and was significantly increased in the *Csf1rko* animals (Fig 7B). The number of glomeruli and so-called nephron endowment is determined during embryonic development [61]. The apparent enrichment in glomeruli in *Csf1rko* can be attributed in large measure to the small size of the kidneys, which are proportional to body weight [17]. By comparison to other tissues, the cluster containing *Csf1r* in kidney data is larger (**Cluster 5;** 166 genes), likely due to greater sensitivity given the complete loss of IBA1-positive cells in this organ [17].

The expression profile of the *Csf1rko* kidney strongly indicates a response to calcium dyshomeostasis and actions of parathyroid hormone (PTH), likely as a response to the loss of osteoclasts in the bone which release calcium into circulation [17,62]. Calcium levels were measured in the serum which confirmed a small but significant decrease in serum calcium (Fig 7C). Fig 7D shows the expression profiles of selected genes in the kidney which are responsible for increasing the reabsorption of calcium in response to this dyshomeostasis. One of the most over-expressed genes in the *Csf1rko* kidney is *Cyp27B1,* encoding the limiting enzyme of vitamin D3 metabolism, 25-hydroxyvitamin D-3 alpha hydroxylase. Vitamin D produced in the kidney acts on the gut where it is the major stimulator of active calcium absorption [63]. PTH responsive calcium channels, TRPV5 and TRPV6 [64,65], were also increased in kidney. The highly-expressed *Adh6* gene, further increased 3-fold in the *Csf1rko*, has been associated with the regulation of vitamin D3 in humans [66]. We speculate that other relatively abundant genes up-regulated in *Csf1rko* kidney (e.g., *Creb3l3, Cyp2e1, Pcsk9, Qrfpr, Tmem252*) may also contribute to calcium homeostasis.

The presence of a substantial macrophage population in the mouse adrenal gland was recognised with the first localisation of F4/80 [67]. Resident F4/80+ macrophages were especially abundant in the zona glomerulosa, the site of mineralocorticoid production. Dolfi *et al.* [68] analysed their diversity in mice by single cell RNA-seq and described sex-specific impacts on the prevalence of a class II MHC+ve subset. Fan *et al.* [69] provided evidence of their function in maintenance of vascular integrity and aldosterone production in adult mice. The histological pattern in the juvenile *Csf1rko* rat suggested an overall hypoplasia of the adrenal cortex compared to the medulla (Fig 7E). Disorganisation was also observed within the cortical layers of the *Csf1rko* rats, the primary difference being a relative hyperplasia of the zona glomerulosa (Fig 7E). Sirius Red staining confirmed that this hyperplasia was not caused by fibrosis. The striations

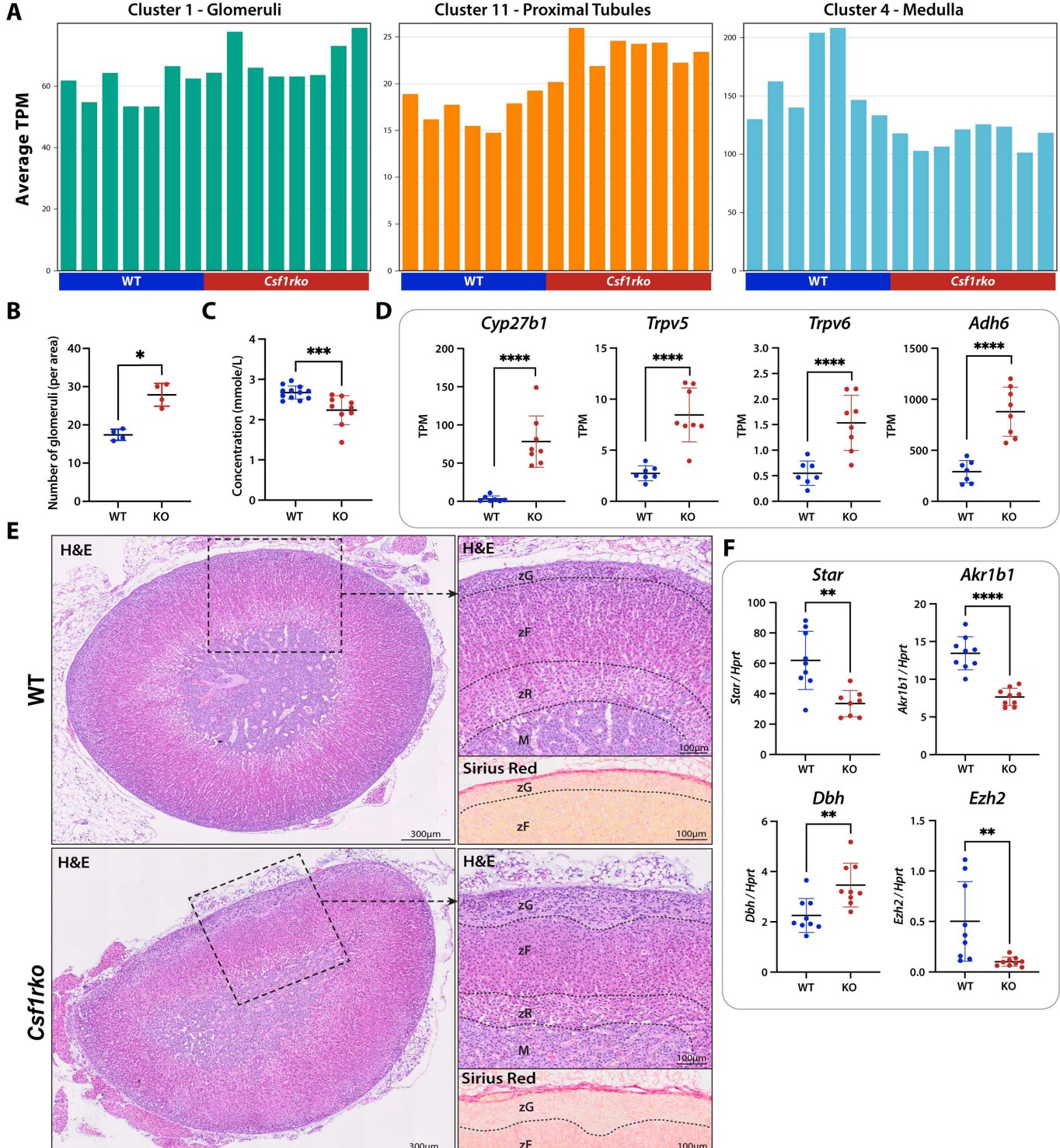

**Fig 7. The effect of the *Csf1rko* on development of the kidney and adrenal gland.** (A) The profiles of Clusters 1, 11 and 4 from Tab A in S6 Table. Y-axis shows the average expression of transcripts in the cluster (TPM); individual columns are biological replicates of the WT and *Csf1rko* tissues as indicated. (B) Quantification of the number of glomeruli contained in the kidney cortex. Each dot represents 1 biological replicate (n= 4 WT; 4 *Csf1rko*); 5 representative areas were sampled and averaged together for each biological replicate. (C) Calcium concentration (in mmole/L) in serum from 3 week male and female WT and *Csf1rko* rats (n = 12 WT; 10 *Csf1rko*). (D) Expression profiles in transcripts per million (TPM) for selected genes from RNA-Seq data (Tab A in S6 Table) of 3wk WT and *Csf1rko* kidney (n = 7 WT; 8 *Csf1rko*). Graphs show the mean ± SD. (E) Representative H&E and Picric Sirius Red images from 3 week WT and *Csf1rko* adrenal glands. Original magnification: 40X. Left image shows a H&E overview of the entire adrenal gland; Scale bar: 300μm. Top right image shows a zoomed in H&E region containing all

layers of the adrenal gland (M: medulla, zR: zona reticularis, zF: zona fasciculata, zG: zona glomerulosa); Scale bar: 100μm. Bottom right image shows Picric Sirius Red staining of the same region of adrenal capsule and zona glomerulosa as the image above; Scale bar: 100μm. (F) qRT–PCR analysis of 3 week WT and *Csf1rko* adrenal glands (n = 9 WT; 9 *Csf1rko*). Graphs show the mean ± SD. Graphs show the mean ± SD. *, P < 0.05; **, P < 0.01; ***, P < 0.001; ****, P < 0.0001.

in the zona fasciculata caused by parallel fenestrated capillaries which are normally present were absent in the *Csf1rko* animals but expression of *Vegfa* and endothelial markers (e.g., *Pecam1, Cdh5, Pvlap*) was unaffected. Consistent with that view, genes in **Cluster 3,** containing the cortex specific steroidogenic acute regulatory protein (*Star*) and aldo-keto reductase family 1, member B1 (*Akr1b1*) were under-represented in the adrenal transcriptome, while medulla-associated genes including dopamine beta-hydroxylase (*Dbh*) were over-represented (**Cluster 10**) in the mutant rats, as confirmed by qRT-PCR (Fig 7F). *Cyp11b2,* encoding the limiting enzyme of aldosterone production, was reduced in females only. The expression of genes within **Cluster 3** was variable in wild-type males and females, possibly related to early gonadal maturity in some of the animals (see below). Greatly-reduced expression of *Ezh2*, encoding an upstream regulator of steroidogenic differentiation [70] in *Csf1rko* adrenal was confirmed by qRT-PCR (Fig 7F), but we did not observe reduced expression of key enzymes involved in steroid biosynthesis (e.g., *Cyp11a1*). As in the kidney, the *Csf1r* coexpression cluster (**Cluster 11**; 107 genes), including class II MHC genes, contains genes that are almost completely absent in the *Csf1rko* adrenal.

## Pituitary, gonads and fat

The pituitary is the second tissue, alongside liver, where there is a clear distinction in transcriptomic profile at weaning between WT and *Csf1rko* (Fig 1A). Our previous analysis compared gene expression in the pituitary in juvenile WT and *Csf1rko* rats [19]. In the present study network analysis was conducted on these pituitary samples. Consistent with the global difference evident in the sample matrix (Fig 1A), *Csf1r* was contained in a large cluster of genes (**Cluster 2**) with a shared pattern of down-regulation in the *Csf1rko* (**Tab A in S7 Table**). As in the liver [17], the *Csf1r*-coexpression cluster includes *Pcna, Mki67* and is enriched for genes associated with cell cycle and growth (as determined by GO term annotation) indicating that postnatal growth of the pituitary is compromised in the *Csf1rko.* Reclustering at higher stringency (MCL inflation value of 3) separates a set of 161 genes more strictly coexpressed with *Csf1r* (Tab B in S7 Table). The set contains no pituitary-specific genes, nor any obvious regulators of cell growth. The pituitary produces multiple hormones that regulate somatic growth and gonadal development and fertility, and we observed changes in the relative expression of these key hormones [19]. Expression of prolactin (*Prl*) was reduced 3-4 fold in the *Csf1rko* in both sexes, and luteinising hormone (*Lhb*) and follicle-stimulating hormone (*Fshb*) were selectively reduced in males. These changes likely contribute to the lack of gonadal development in both sexes.

The core set of macrophage-expressed transcripts detected in other tissues was also readily detected in WT testis, and almost undetectable in the *Csf1rko*; they mainly fall within **Cluster 6 (Tab C in S7 Table).** However, others, including *Adgre1* and *Aif1*, fall within a much larger cluster (**Cluster 2**) that was selectively expressed in 3/5 WT males and is highly-enriched in genes associated with spermatogenesis. We infer that these three WT animals were more advanced in sexual maturation. Interestingly, this cluster includes the macrophage growth factor and CSF1R ligand, *Il34.* Similarly, in the ovary, the set of macrophage-related genes, also almost completely CSF1R-dependent, forms part of much larger cluster (**Cluster 5**) of genes that are associated with ovarian development (Tab D in S7 Table).

One of the more obvious macroscopic phenotypes in *Csf1rko* rats is a severe loss of visceral fat. This meant that we were only able to obtain sufficient tissue for mRNA isolation and profiling from 2 male and 1 female *Csf1rko* rats compared to 2 female and 3 male controls. The

expression of adipocyte-specific genes (*Adig, Adipoq*) was variable in both control and *Csf1rko* rats and network analysis produced clusters that were specific to individual samples (Tab E in S7 Table). However, we identified a set of 200 genes that were consistently expressed across the 5 controls and reduced > 5-fold in *Csf1rko* adipose (Tab F in S7 Table). The CSF1R-dependent gene set includes two known regulators of mouse adipose development, *Il18* and *Pdgfc* [71,72] supporting the view that adipose hypoplasia is due at least in part to local macrophage depletion.

## BMT-derived macrophages acquire WT tissue macrophage expression signature in most tissues

We previously demonstrated that the severe phenotypes in *Csf1rko* rats can be reversed by bone marrow transfer (BMT) at weaning, which was associated with restoration of circulating IGF1 and all tissue macrophage populations [17,18]. Whole mount images in S4 Fig confirm the effective repopulation of these tissue macrophages following BM, including complete repopulation of adipose tissue which is restored to normal levels in BMT recipients.

The three subunits of C1Q encoded by *C1qa, C1qb* and *C1qc* were highly-expressed in most tissues and absolutely CSF1R-dependent. Their expression in mice is believed to be largely restricted to macrophages, regulated by lineage specific transcription factors PU.1 (Spi1) and MAFB [73,74]. C1Q deficiency in *C1qa*[-/-] mice can be restored by BMT [75], a clinically important observation in view of the autoimmunity associated with C1Q deficiency. We used a recently-developed quantitative ELISA to measure circulating C1Q [76]. Consistent with the complete dependence on macrophages as the source, C1Q was undetectable in *Csf1rko* rat serum. Following BMT, C1Q was rapidly restored and reached WT levels within 2 weeks (Fig 8A).

In the brain, the FACS profile of isolated microglia was clearly different between WT and BMT recipients. The absolute abundance of microglia was restored by BMT, as shown previously by IBA1 staining [17] and they were of donor origin (*Csf1r*-mApple+). However, the cells isolated from BMT recipients had higher CD45 compared to controls (Fig 8B). We compared gene expression profiles in brains of WT and long term BMT (12 weeks) recipients. The expression data, network analysis and differential expression analysis are presented in Tabs A-C in S8 Table. Consistent with FACS and immunohistochemical analysis, *Ptprc* (CD45) was higher, *Itgam* (CD11b) was lower and *Aif1* (IBA1) mRNA levels were identical in BMT compared to control brains. Other myeloid-associated genes were restored to different extents as summarised in Table 1 and shown in Fig 8C-E . Homeostatic microglial genes such as *P2ry12* and *Tmem119* were restored to around 20% of WT (Fig 8C), whereas genes expressed by so-called damage-associated microglia [77] such as *Clec7a* were substantially over-expressed in the BMT brain (Fig 8D). Other macrophage markers such as *Adgre1, Aif1,* and *C1qa*, and markers specific to border-associated macrophages (BAMs) including *Cd163* were restored to WT levels (Fig 8E). Notwithstanding the apparent presence in BMT recipients of a microglial phenotype associated with neuropathology, there was no evidence of pathology. Indeed, markers of stress/pathology that were elevated in 3 week old *Csf1rko* brains [19], including *Gfap, C4b, Rbm3* and *Mt1/2a,* were normalised in BMT recipients, perhaps suggesting that the damage-associated phenotype is involved in repair rather than injury.

The major CD172a+/CD11b+ resident peritoneal macrophage population is completely absent in *Csf1rko* rats, and the total cellular yield from peritoneal lavage is 1-2% of a WT rat [15]. In BMT recipients, the CD172[+]/CD11b[+] population was restored entirely from cells of donor (*Csf1r*-mApple[+]) origin [18]. To investigate peritoneal macrophage repopulation at a molecular level, we profiled peritoneal lavage cells from WT and *Csf1rko* BMT recipients. Tab A in S9 Table compares the profiles of the two populations and cluster analysis is shown in Tab B in S9 Table. The core signature of the resident peritoneal macrophages was identical

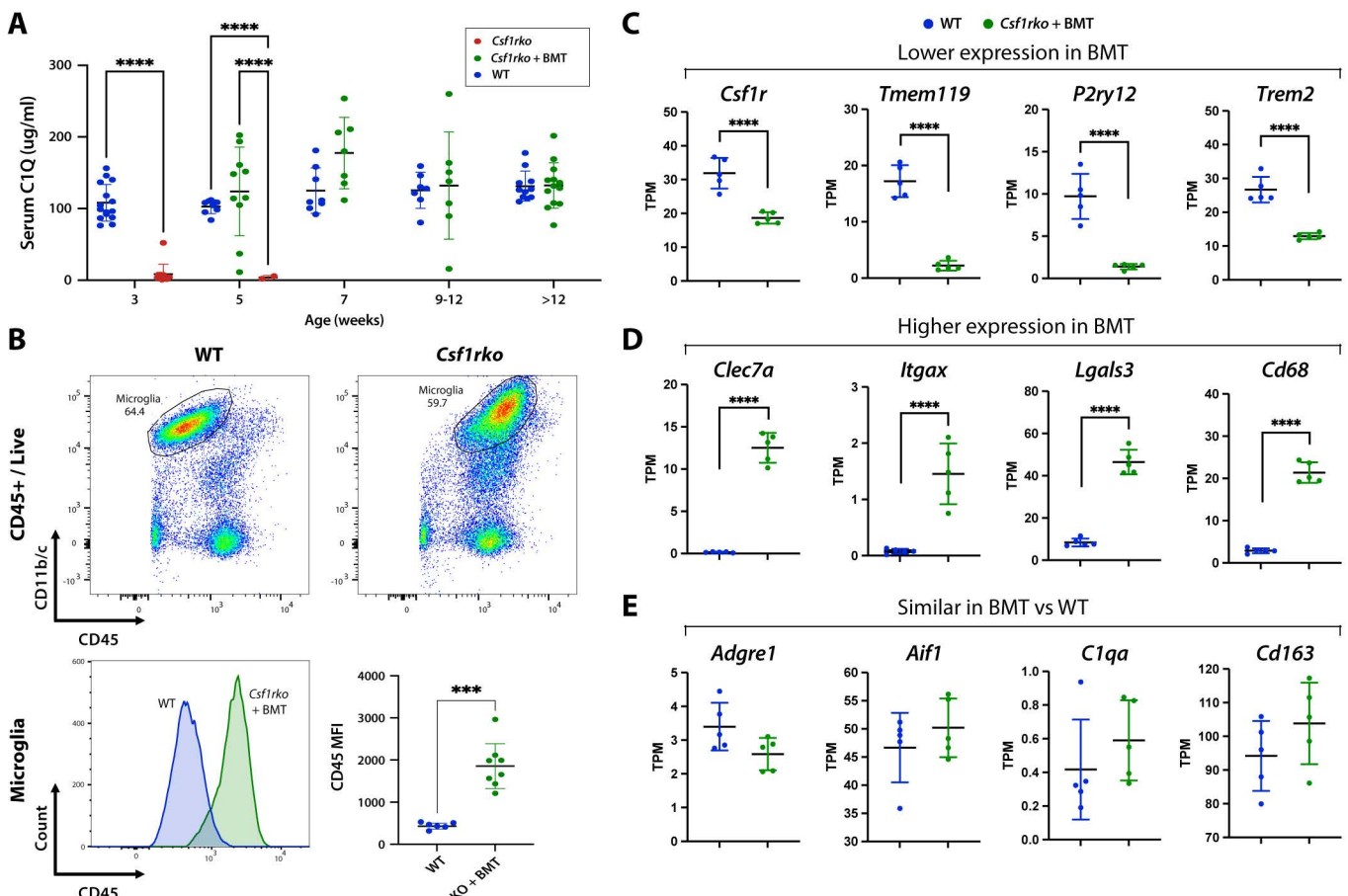

**Fig 8. Phenotypic reversal and repopulation of tissue macrophages in the *Csf1rko* following WT bone marrow cell transfer (BMT).** *Csf1rko* rats received an intraperitoneal injection of WT bone marrow cells at 3 weeks of age. Measurement of C1Q levels in WT, *Csf1rko*, and *Csf1rko* BMT recipient serum by quantitative sandwich ELISA. Each point represents data from a single animal. (**B**) Representative flow cytometry plots of disaggregated whole brains from WT and *Csf1rko* BMT recipients and barplot showing the Mean Fluorescence Intensity (MFI) of CD45 within the microglial population (n = 6 WT; 8 *Csf1rko* + BMT. Ages range from 8-35 weeks). (**C–E**) Expression profiles in transcripts per million (TPM) for selected genes from RNA–Seq data of brains from WT and *Csf1rko* BMT recipients. Barplots show genes that showed (**C**) lower expression in BMT animals (**D**) higher expression in BMT animals or (**E**) similar expression in BMT animals vs WT animals. Primary data is in S9 Table. Graphs show the mean ± SD.

**Table 1. Differential restoration of myeloid-associated genes in the brain of WT and BMT-rescued *Csf1rko* rats.**

| Lower expression in BMT | Similar in BMT vs WT | Higher expression in BMT |
|---|---|---|
| *Alox5, Ccr5, Cd7, Cd180, Csf1r, Ctss, C3, Fcrla, Gpr34, Gpr84, Havcr2, Ltc4s, Plac8, P2ry12, P2ry13, Selplg, Siglec5, Slc2a5, Tlr2, Tmem119, Trem2* | *Adgre1, Aif1, Axl, Cd4, Cd74, Cd163, Csf3r, Cx3cr1, C1qa/b, C5ar1/2, Fcgr2b, Hexb, Irf8, Itgam, Laptm5, Mrc1, Nckap1l, Siglec1, Slco2b1, Spi1, Tifab, Tyrobp,* | *Batf, Batf3, Cd6, Cd14, Cd68, Clec4a1, Clec7a, Clec10a, Clec12a, Cxcl13, Cybb, Fcgr1a, Fcgr3a, Folr2, Gpnmb, Itgax, Lgals3, Lyz2, Mafb, Msr1, Ms4a7, Plau, Tnfsf13b* |

between the WT and BMT populations, including *C1qa/b/c*, highly-expressed surface markers (e.g., *Adgre1, Cd4, Cd81, Clec10a, Fcgr1a, Icam2, Itgam, Lgals9, Lyve1, Siglec5, Trem2*) [24,78,79] and a complex transcription factor profile that includes *Bhlhe40, Cebpa/b/d/e, Gata6, Hif1a, Irf1/2/3/5/7/8/9, Klf2/4, Maf1, Mafb, Rara, Stat1/2/3/5a/5b/6, Spi1*, and *Tfec* (examples of these genes shown in S5A Fig) [79–81]. A small cluster of less abundant genes

with reduced expression in the BMT, including *Cx3cr1,* may reflect the continued lack of non-classical monocytes following BMT. Lymphocyte markers(*Cd19, Cd3*) were also low but consistent between WT and *Csf1rko* BMT animals. We also detected a clear mast cell cluster containing mast cell proteases (e.g., *Cma1* (chymase), *Cpa3, Mcpt1l1*) and tryptases (*Tpsab1*), along with *Csf1*. The rat peritoneal cavity was reported to contain a population of tissue resident mast cells as much as 25% of total lavage [82], although detection may vary due to the sensitivity of mast cells to degranulation. We also confirmed the presence of abundant toluidine blue positive cells in rat peritoneal lavage (S6 Fig). Together the data suggest BMT recipients establish a resident peritoneal macrophage population that is indistinguishable from WT, and proportionate to other resident populations, notably mast cells, which may be a significant source of CSF1.

We also profiled the lung and liver of BMT recipients and age matched controls. In the lung, the signature of CSF1R-dependent macrophages found in the analysis of the 3 week WT and Csf1rko animals (Fig 3A) (e.g. *Adgre1, Cd163, C1qa, Folr2*) was restored (S5B Fig) and there was no longer evidence of the neutrophilia (e.g., *Mmp8/9, Ngp, S100a8/9*) seen in juveniles (Tabs C and D in S9 Table). Similarly, in the liver, the relative expression of all CSF1R-dependent genes identified at 3 weeks of age, including Kupffer cell-specific genes (e.g., *Cd5l, Cd163, Clec4f, Marco, Vsig4*) and genes involved in lipid metabolism [17], was indistinguishable between controls and long term BMT recipients (S5C Fig and **Tabs E and F in** S9 Table). qRT-PCR analysis of the adrenal glands of BMT recipients confirmed the restoration of *Csf1r* and *Ezh2* expression, along with restoration of the cortex specific genes (*Akr1b1, Star*) and medulla specific gene *Dbh* (S5D Fig).

Our previous analysis of repopulation following BMT using the *Csf1r*-mApple transgene as a marker revealed clusters of donor-derived cells in distant sites suggesting seeding by highly-proliferative progenitors [18]. CD209B provides a unique marker to assess local repopulation and specialisation in lymphoid tissue. Accordingly, we localised CD209B in spleen and LN at various times post BMT. S7 Fig shows complete repopulation of CD209B-positive macrophages. At earlier times we observed clusters of positive cells in both locations suggesting that local proliferative expansion and adaptation may precede migration to fill the vacant niche.

## Discussion

### The interaction between liver, pituitary and bone in Csf1r-dependent control of somatic growth

*Csf1rko* rats have a severe defect in postnatal somatic growth including the loss of the post-pubertal differential growth in males. In some respects, the phenotype resembles the postnatal growth defects associated with mutations in ligands and receptors in the growth hormone/insulin like growth factor axis (GH/IGF1) [83,84]. At least at the mRNA level, *Csf1rko* rats are not absolutely deficient in expression of growth hormone (GH) mRNA (*Gh1* gene) in pituitary and GH protein is detected in the circulation [17]. Prolactin (PRL; *Prl* gene), which was reduced by 70-80% in the pituitary in 3 week old *Csf1rko* rats, has also been implicated in control of postnatal liver growth in mice [85]. The prolactin receptor (*Prlr*) is highly-expressed in juvenile rat liver, with a significant female bias which was selectively abolished in the *Csf1rko* [17]. However, expression of the GH/PRL target gene *Socs2* in liver was not affected. Analysis of various liver-specific conditional mutations in mice indicates that the liver is the main source of circulating IGF1, but the loss of hepatic expression does not mimic a complete *Igf1* knockout [83,84,86]. In the rat *Csf1rko,* circulating IGF1 was profoundly reduced, *Igf1* and *Ghr* in liver were reduced by around 60%, and *Igfals*, encoding an essential serum IGF1

binding protein, by 80% [17]. Although CSF1-stimulated macrophages express high levels of *Igf1* mRNA *in vitro,* the absence of any significant effect of the *Csf1rko* on *Igf1* mRNA in tissues other than pituitary in the current study indicates they are not a major extra-hepatic source in rats. The lack of hepatic IGF1 in transgenic mice with liver-specific conditional deletion may be compensated in part by up-regulation of IGF2 [87]. In keeping with that view, *Igf2* mRNA was increased significantly in *Csf1rko* muscle. Although IGF1 is proposed to exert feedback regulation on GH, there was no detectable compensatory increase in *Gh1* mRNA in the pituitary. Another feature of the liver transcriptome of the *Csf1rko* is the marked increase in expression of *Igfbp1* and *Igfbp2* [17]. Hepatic over-expression of these IGF binding proteins is sufficient to inhibit somatic growth [84,86]. IGFBP1 has been attributed a function as a liver-specific endocrine regulator of osteoclastogenesis [88].

Although there is evidence for regulation of osteoclast (OCL) function by IGF1 [83,84] the complete loss of OCL and resident bone macrophages in the *Csf1rko* [62] and consequent osteopetrosis is specific to the mutation. Delayed skeletal calcification is evident in *Csf1rko* newborns [17]. Paradoxically, serum calcium and phosphate were reduced in the adult *Csf1rko* rats [15], which we confirmed in juveniles. This is likely a consequence of elevated parathyroid hormone (PTH). The primary function of PTH is to maintain calcium homeostasis [89]. In response to reduced serum calcium, PTH released by the parathyroid acts to induce RANKL in osteoblasts to promote osteoclastic bone resorption releasing calcium, which cannot occur in the OCL-deficient *Csf1rko* rats. We detected the signature of hyper-parathyroidism in the kidney, with 17-fold induction of *Cyp27B1,* the limiting enzyme for vitamin D synthesis, and the tubular calcium transporter *Trpv5*. Increased serum vitamin D3 has also been detected in osteopetrotic *Csf1*[tl/tl] rats [90]. Similarly, in the intestine, we observed induction of the inducible calcium transporter, *Trpv6* [91]. Although kidney and bone are regarded as the main targets of PTH, the receptor *Pthr1*, was highly-expressed in juvenile rat liver. PTH activates cyclic AMP production in hepatocytes and cyclic AMP induces *Igfbp1* transcription [92,93]. Dysregulation of osteoblast differentiation and the lack of osteoclasts is also likely to alter the release and post-translation modification of the hormone-like mediator, osteocalcin [94]. Osteocalcin (*Bglap* gene) was reduced in bone and increased in serum in *Csf1*[tl/tl] rats [90]. In summary, there is likely a two-way interaction between bone and liver that contributes to postnatal growth retardation in *Csf1rko* rats, linked to calcium dyshomeostasis.

Deficiency in thyroid hormone, an essential regulator of liver maturation, skeletal maturation, postnatal somatic growth and lipid/cholesterol metabolism [95–97], is another likely contributor to impaired growth. The signature of thyroid hormone insufficiency is evident in the gene expression profile of the *Csf1rko* at weaning, with known thyroid hormone-responsive genes (e.g., *Dio1, Fasn, Hmgcs1, SerpinA6, Srd5a1, Thrsp*) [98] amongst the most deficient. Thyroid hormone deficiency leads to hepatic steatosis [98] which is a feature of *Csf1rko* rats that is reversed by BMT [17]. Down-regulation of SERPINA6, the major serum glucocorticoid-binding protein, and genes for glucocorticoid metabolising enzymes (*Srd5a1, Hsd11b1*) may contribute in turn to dysregulation of the stress response and sex-specific differences in development of the adrenal gland [99,100]. We observed increased expression of stress-responsive genes including metallothioneins (*Mt1, Mt2a*) [101] and *Rbm3* [102] in every organ studied including the brain.

The *Csf1rko* rats are already severely growth retarded prior to weaning and the start of puberty. Puberty in rats is considered to begin at around 4-5 weeks [99]. In our cohort at 3 weeks, sexually dimorphic gene expression in the liver [99] was not yet detectable but based upon transcriptomic analysis a subset of the wild-type animals already showed evidence of the onset of sexual maturity in testis and ovary that was absent in any of the *Csf1rko*. The knock-out animals fail to show a pubertal growth surge, and the male growth advantage is absent

[17]. The *Csf1rko* animals remain IGF1-deficient well beyond the normal age of puberty [17]. Given the clear impacts on gonadotroph differentiation in the pituitary at 3 weeks, it is likely that the failure of gonadal development is at least partly extrinsic, rather than a direct consequence of the local depletion of macrophages in the gonads.

Amongst all the tissues profiled, the liver has the clearest pattern of down-regulation of cell cycle-related transcripts indicating reduced hepatocyte proliferation, as well as a profound delay in functional maturation [17]. Other studies in our lab have demonstrated that exogenous CSF1 treatment can promote the growth of the liver and identified CSF1 as a key component of the hepatostat which maintains a relatively constant liver-body weight ratio [103–106]. Analysis of a liver-specific conditional *Jun* deletion [107] in mice suggests that reduced growth in the liver can be causally linked to reduced somatic growth. However, despite the reduced liver growth, the liver:body weight ratio is not affected in *Csf1rko* rats [17].

## The functions of Csf1r-dependent macrophages in hematopoiesis and organ maturation are largely redundant

Microglia have been attributed numerous functions in postnatal development in the brain [77] but their complete absence in *Csf1rko* rats or in a hypomorphic mouse *Csf1r* mutant has surprisingly little impact on region-specific gene expression [19,108]. In the bone marrow, aside from OCL and osteomac deficiency, the *Csf1rko* rat lacks a resident macrophage population [62] that normally forms the centre of hematopoietic islands [109,110]. Fetal liver macrophages also provide a niche for the development of primitive erythroid cells, but these cells appear absent in the *Csf1rko* fetal liver without any apparent consequences for the generation of red blood cells [16]. Recent evidence in mice indicates that erythroblastic islands, containing a central macrophage surrounded by maturing erythroid cells, also provide a niche for terminal granulopoiesis [110] and maintain a balance between myeloid and erythroid production. The *Csf1rko* rat has an imbalance in favour of granulocytopoiesis over erythroid-megakaryocyte and B lymphocyte production, but the animals are not deficient in either red blood cells or platelets [15]. We suggest that reduced production of both red cells and platelets is linked to reduced clearance by tissue macrophages and some form of feedback control. We noted reduced detection of hemoglobin transcripts in every organ, especially the liver, which is likely a consequence of reduced numbers of erythrophagocytic macrophages, as well as >30-fold increase in expression of *Hamp* (hepcidin), the homeostatic regulator of iron transporters, in the liver. The latter observation is consistent with a regulatory pathway linking Kupffer cell erythrophagocytosis and hepcidin expression [111]. Similarly, CSF1 administration causes a transient thrombocytopenia, attributed to increased platelet clearance, which is resolved by an increase in platelet production [112]. If altered clearance of red cells and platelets is the mechanism that maintains homeostasis in the blood, then the imbalance between erythroid and myeloid fates in *Csf1rko* marrow is extrinsic and the function of resident marrow macrophages is entirely redundant.

Unlike the extensive effects of the *Csf1rko* on the transcriptome of pituitary and liver, the impacts on tissue-specific gene expression in other major organs were subtle. The most obvious effect was the loss of macrophage-associated genes in each tissue, discussed below, but we also identified examples of developmental delay in multiple organs and subtle impacts on the peripheral nervous system and myelination. The severe effect of the *Csf1rko* on adipose development may reflect a role for resident macrophages as a source of growth factors, PDGFC and IL18, but it is likely also to reflect the complete loss of circulating IGF1 [113]. Indeed, given the systemic impacts of the *Csf1rko,* and the lack of clear dysregulation of known transcription factors and tissue-specific growth factors, it is really impossible to attribute any of these changes specifically to local macrophage roles.

## Identification of a generic signature of tissue resident macrophages

By correlation analysis of the entire dataset we identified a generic signature of 48 genes that are tightly correlated across non-lymphoid tissues which defines CSF1R-dependent macrophages in the rat. That signature includes C1Q, which was validated at the protein level and provided a sensitive biomarker of repopulation of the mononuclear phagocyte system (MPS) following BMT (Fig 8A). Surprisingly, the only transcription factor mRNA in this cluster was *Irf5*. IRF5 is commonly considered a marker of inflammatory macrophages [114] but it is constitutively expressed by diverse resident macrophage populations in mice [24] and rats [31]. In mice, Dick *et al.* [115] identified *Timd4/Lyve1/Folr2* (TLF) as a shared transcriptomic signature of yolk sac-derived self-renewing resident macrophages in every major organ of C57Bl6/J mice based upon scRNA-seq analysis. However, lineage-trace models have not been applied to other mouse strains or outbred mice, or other species [116]. Furthermore, it is clear that tissue disaggregation approaches do not generate a representative sample of resident tissue macrophages, many of which are fragmented, contaminated with unrelated cell types or activated in the isolation process [24,28,116,117]. For example, this artefact has likely confounded the isolation and characterisation of interstitial macrophages in the mouse lung [24,28]. Recently, Ural *et al.* [118] compared RNA-seq data from isolated CSF1/CSF1R-dependent mouse interstitial lung macrophages (IM) to alveolar macrophages. Of the IM markers they defined, only a small subset (*C1q, Adgre1, Clec10a, Cd163*, Class II MHC) overlap the CSF1R-dependent gene cluster in the rat (Cluster11, S2 Table) whereas *Siglec1* (CD169) and *Mgl2* are undetectable and *Mrc1* (CD206) is not CSF1R-dependent.

Of the TLF signature, only *Folr2* translates to rat. The CSF1R-dependent cluster does not include *Timd4,* which is almost undetectable in tissues other than CSF1R-dependent expression in spleen and lymph node (in common with mouse, see data on BioGPS.org). This species difference may relate to promoter variation, since the rat *Timd4* gene appears to lack a TATA box that is present in the mouse proximal promoter (see www.ensembl.org). The functional consequence of the absence of TIM4 may be compensated in some tissues by macrophage expression of related genes, *Timd2* in liver and *Havcr2* (TIM3) in multiple organs including brain. *Lyve1* was readily detected in most tissues but with the exception of brain regions and adipose, it was not significantly reduced by the *Csf1rko* suggesting the main location of expression is in lymphatic vessels and *inter alia*, that their development is macrophage independent.

The F4/80 antigen encoded by the *Adgre1* gene was defined as a marker of interstitial macrophages in mice in the 1980s [119] and has been used extensively since [120]. The rapidly evolving *Adgre1* gene is expressed by macrophages in all mammalian species [121]. *Adgre1* is CSF1R-dependent in every rat tissue suggesting that the expression is similar to mouse. We are currently confirming this prediction using a recently available anti-rat ADGRE1 antibody [122]. CD4 has been identified as a marker of subsets of tissue macrophages in mouse intestine but is not widely expressed by mouse macrophages [24]. By contrast, in rats [123] and humans [124] CD4 protein is expressed by monocytes and macrophages and *Cd4* forms part of the core macrophage signature in our analysis.

One prominent feature of the CSF1R-dependent signature was the presence of mRNAs encoding multiple C type lectins of the *Clec4a* family, as well as *Clec10a* and *Cd33,* which have been associated with antigen uptake and antigen-presenting cell function in mice. These proteins have been referred to collectively as dendritic cell immunoreceptors, but this nomenclature is misleading and is not consistent with their more widespread expression in mouse MPS cells [24]. The expression signature of classical DC in the mouse [24] including *Flt3, Clec9a, Ly75, Ccr7, Itgae, Batf3, Xcr1* and *Ztbt46* was detectable in total RNA-seq data in every rat tissue we profiled but was unaffected by the *Csf1rko.* By contrast to mice, CSF1R-dependent

macrophages, including resident peritoneal macrophages as shown here, express class II MHC, and accordingly detection of *Cd74,* the transcriptional regulator *Ciita* and class II MHC genes was CSF1R-dependent in most tissues. Residual CSF1R-independent MHCII expression in some tissues may be associated with the DC populations as well as B cells. Unlike the *Clec4a* family, *Clec4b2* was not CSF1R-dependent. DA rats actually have a loss-of-function mutation in this gene linked to genetic susceptibility to auto-immune disease [125].

Notwithstanding the abundant literature on tissue-specific macrophage adaptation [5,6,126,127] we found few examples of tissue-specific CSF1R-dependent markers outside the liver and lymphoid organs. This conclusion is consistent with our published meta-analysis of profiles of isolated mouse macrophage populations [24]. Analysis of a comprehensive rat expression atlas that included isolated macrophage populations identified a set of genes that was strongly-enriched in macrophages compared to all tissues [23]. S10 Table compares the set of CSF1R-dependent genes from this previous study with the lists of CSF1R-dependent genes in this study, including the global macrophage-enriched cluster [23], the microglial signature [19] and the set of CSF1R-dependent genes in tissues (kidney, heart, muscle, adipose, adrenal, pituitary, ovary) in which IBA1[+] cells were strongly depleted. Many of the genes within this list encode known endocytic receptors and macrophage markers and the evidence that resident macrophages are the dominant site of expression in tissues is not surprising. Others are less obvious and provide new insights. For example, leukotriene C4 synthase (*Ltc4s*), traditionally linked to mast cells and susceptibility to asthma [128] was highly-expressed and almost completely CSF1R-dependent in all tissues including the lymph node, lung and brain. A rare human LTC4S deficiency was associated with severe muscular hypotonia, psychomotor retardation and failure to thrive [129]. *Ltc4s* knockout mice have attenuated innate and acquired immunity [130] and develop spontaneous emphysema [131]. Similarly, the macrophage-restricted expression of *Tbxas1* extends previous evidence [132] that these cells are the major source of the vasoactive arachidonic acid metabolite, thromboxane A2.

## Csf1r-dependent macrophages in lymphoid tissues are specialised for antigen capture

Lymph nodes in mice contain multiple resident MPS cell populations including classical dendritic cells that depend upon FLT3-ligand, T cell zone macrophages, tingible body macrophages in germinal centres, subcapsular sinus macrophages (CD169+) and medullary sinus macrophages (CD209B+) [133–136]. The latter two populations are adapted to capture antigens, including viral and microbial pathogens, entering via the lymphatics and they are functionally related to the marginal zone and marginal zone metallophilic macrophages of the spleen. Both populations in spleen and LN are depleted by anti-CSF1R treatment [135–137]. Transcriptomic analysis of lymph node and splenic marginal zone macrophages has been compromised by their fragmentation during tissue disaggregation [117,136,138]. Identification of significant differentially-expressed genes between wild type and *Csf1rko* lymph nodes (Fig 4B) provides an alternative way to extract functional information about LN macrophages. We confirmed the CSF1R-dependent high expression of the transcription factor *Mafb* and signature genes such as *Adgre1, C1qa/b/c, Timd4* and *Folr2*, alongside those encoding numerous lectins including all 7 members of the rat CD209 cluster (*Cd209, Cd209a/b/c/d/e/f*), *Siglec1/5/10, Cd33* (aka *Siglec3*), *Clec4a/a1/a3, Clec4m* (aka L-SIGN), *Clec10a, Mrc1* and other endocytic receptors (*Havcr1, Fcgr1a, Stab1*). We confirmed the complete loss of CD209B (SIGNR1) expressing macrophages in both LN and spleen of *Csf1rko* rats and identified a novel CSF1R-dependent CD209B+ population in the thymus (Fig 4A). *IL22ra2,* which encodes a decoy receptor and inhibitor of IL22, was amongst the most highly-expressed

CSF1R-dependent genes and almost entirely lost in *Csf1rko* lymph nodes as seen previously in the spleen [15]. Jinnohara *et al.* [139] reported the expression of *Il22ra2* in sub-epithelial CD11b+ DC in mouse Peyer's patches and decreased antigen uptake by M cells in *Il22ra2-/-* mice. They did not detect *Il22ra2* in isolated DC from spleen or lymph node. However, they did not examine expression in macrophages *in situ*. Unexpectedly, in rat LN we also observed CSF1R-dependent expression of *P2ry12/13* and *Tmem119*, commonly considered microglial markers. In common with previous analysis in spleen [15], the macrophage-enriched transcription factor, *Tfec* [140,141] appears as a candidate transcriptional regulator in lymph node macrophages. Highlighting the power of our approach, few of the CSF1R-dependent genes we identified in rat LN were detected in scRNA-seq analysis of mouse LN [135] and some doubt would be cast upon identification of subpopulations in that study.

We noted previously a population of IBA1+ cells within T cell areas of LN and spleen that were unaffected by the *Csf1rko* and were not replaced by donor cells in BMT. The CSF1R-independent MPS population in LN may include T cell zone macrophages [133], expressing *Aif1, Itgam* (Cd11b), *Cx3cr1, Fcgr1a* and *Mertk*, as well as FLT3-dependent classical dendritic cells. *Flt3lg* was highly-expressed in LN, and definitive markers of cDC1 including *Clec9a, Flt3, Ly75, Xcr1,* and *Zbtb46* [24] were not affected by the *Csf1rko*. The T cell zone macrophages may, like those of the lung, be maintained by CSF2. Genes encoding the CSF2 receptor (*Csf2ra, Csf2rb*) were highly-expressed in rat LN and unaffected by the *Csf1rko*, and the ligand (*Csf2*) was significantly up-regulated.

## Complete rescue and maintenance of resident macrophages in Csf1rko rats can be achieved by adoptive transfer of wild-type bone marrow cells.

The analysis of peritoneal macrophages (S9 Table) repopulated following BMT demonstrated that tissue-specific adaptation to the cavity niche can be fully recapitulated by bone marrow-derived cells of donor origin. Similarly, in the liver, the relative expression of macrophage-expressed transcripts including KC-specific genes was precisely restored and the transcriptomic profiles of WT and BMT livers were indistinguishable. We have shown elsewhere that the restoration of normal ossification and bone marrow architecture involves progressive infiltration by donor-derived macrophages prior to regeneration of osteoclasts [62]. Remarkably, the hematopoietic island macrophages, osteomacs and osteoclasts in BMT recipients are of donor origin whereas monocytic progenitor cells remain CSF1-unresponsive [17,18,62]. In the case of the lung, although alveolar macrophages express high levels of *Csf1r* mRNA [23,31], they are not CSF1R-dependent; but are nevertheless replaced by donor *Csf1r*-mApple cells in the BMT [17,18]. The current analysis indicates that their gene expression profile adapted appropriately to the airway environment and was sufficient to reverse the emphysema-like pathology that is the major cause of morbidity in *Csf1rko* rats. This observation may have implications for the treatment of bronchopulmonary dysplasia that is a common morbidity in pre-term infants [142]. The repopulation of the *Csf1rko* airways as well as interstitial sites by donor-derived macrophages indicated that CSF1R signalling contributes to homeostasis. Consistent with that conclusion, *Csf1* deficiency is associated with emphysema in mice [26] and exacerbates the lung pathology observed in *Csf2*-deficient animals [143].

Only in the brain was there evidence that donor bone marrow-derived cells fail to fully recapitulate the gene expression profile of microglia, in keeping with analysis in *Csf1rko* mice rescued by neonatal BMT [14]. Despite this difference, even 35 weeks post BMT we found no evidence of the extreme neurodevelopmental abnormalities seen in patients with homozygous *CSF1R* mutation [11,12] nor the age-dependent brain pathology (demyelination, thalamic calcification) described in microglia-deficient mice [55,144,145].

Our previous study provided evidence that repopulation of the MPS of *Csf1rko* rats by WT donor cells involves migration and tissue infiltration by relatively immature IBA1[-ve], *Csf1r*-mApple[+ve] cells which proliferate locally [18]. Both the initial migration/proliferation and the final relative density are presumably dictated by the local and systemic availability of CSF1R ligands, CSF1 and IL34. Here we identified an intermediate stage in repopulation of CD209B+ macrophages in spleen and LN, with development of clusters of positive cells detected in advance of restoration of marginal zone/medullary sinus populations (S7 Fig). Circulating CSF1 returns to baseline levels as tissue macrophage populations are restored [18]. Once repopulated, the tissue macrophages must be maintained without input from monocytes, which remain deficient and negative for CSF1R. The nature of the migrating progenitor remains unclear. The analysis of the relationship to macrophage lineage committed progenitors [146] awaits generation of key antibodies against KIT and FLT3 which are not currently available for the rat.

## Conclusions

In overview, this comprehensive analysis of the *Csf1rko* rat indicates that many proposed functions of macrophages in postnatal development [6] are at least partly redundant. The phenotypic consequences of macrophage deficiency are associated with complex dysregulation of a pituitary-bone-liver hormonal axis. Within that complex framework, the clearest non-redundant function of CSF1R-dependent cells is the osteoclastic resorption of bone, but osteoclast depletion in neonatal mice has no effect on somatic growth [147] and *Csf1*-deficient rats, which lack osteoclasts, are not severely growth inhibited [32]. The most remarkable finding in this model is that all of the deficiencies that occur in the *Csf1rko* rat prior to weaning are reversible. Against that background, we have shown previously that CSF1 administration to neonatal rats and infant pigs is well-tolerated and can drive expansion of resident tissue macrophage populations [105,106]. We therefore suggest that CSF1 has potential for the treatment of innate immune deficiency associated with prematurity.

## Materials and methods

### Ethics statement

All animal experiments and protocols were approved by The University of Queensland Health Sciences Ethics Committee (2022/AE000191). Rats were bred and maintained at The University of Queensland (UQ). *Csf1rko* rats were housed with WT littermates and provided a supplemental diet comprising wet mashed standard chow along with veterinary powdered milk nutritional supplement (Di-Vetelact, Sydney, Australia).

### Bone marrow transfer (BMT)

Bone marrow was collected from femurs and tibias of mApple + WT rats by flushing through a 40μm filter using a 21G needle. Cells were washed with PBS and resuspended in saline solution at 20x10⁶ cells/100μL. 100μL was injected intraperitoneally into 3wk-old *Csf1rko* recipient rats. Recipients were maintained on the supplemental diet throughout the experiment and were co-housed with WT littermates.

### RNA purification

RNA was extracted from whole tissue using TRI Reagent (Sigma-Aldrich, MO, USA), according to the manufacturer's instructions. Each extraction used ~100mg of tissue in 1mL of reagent. RNA was digested with DNAse I amplification grade (Thermo Fisher Scientific, MA, USA) in order to remove genomic DNA contamination. RNA quantity was measured on a

Nanodrop Microvolume Spectrophotometer (ThermoFisher, MA, USA), while RNA integrity was measured on an Agilent 2200 Tapestation System (Agilent, CA, USA).

## qRT-PCR

Quantification of gene expression was conducted using quantitative reverse transcription PCR (qRT–PCR). 1ug of total RNA was used to synthesise cDNA, using the SensiFAST cDNA Synthesis Kit (Meridian Bioscience, OH, USA). RT-PCR was performed using the SYBR Green PCR Master Mix (Thermo Fisher Scientific, MA, USA) and was run on either an Applied Biosystems QuantStudio or ViiA7 real-time PCR system (Themo Fisher Scientific, MA, USA). Gene expression relative to the housekeeper gene (*Hprt*) was calculated using the deltaCT method.

## Library preparation and sequencing

Library preparation and sequencing was performed at the University of Queensland Sequencing Facility (University of Queensland, Brisbane, Australia). Bar-coded RNA-Sequencing (RNA–Seq) libraries were generated using the Illumina Stranded mRNA Library Prep Ligation kit (Illumina, CA, USA) and IDT for Illumina RNA UD Indexes. The bar coded RNA libraries were pooled in equimolar ratios prior to sequencing using the Illumina NovaSeq 6000. Paired-end 102 bp reads were generated using either a NovaSeq 6000 S1 reagent kit v1.5 (200 cycles), SP reagent kit v1.5 (200 cycles) or an S2 reagent kit v1.5 (200 cycles). After sequencing, fastq files were created using bcl2fastq2 (v2.20.0.422). Raw sequencing data (in the form of.fastq files) was provided by the sequencing facility for further analysis.

## Quality control and read pre-processing

Raw reads were pre-processed using fastp v0.23.2 [148] using previously described parameters [17]. FastQC [149] was used on pre and post-trimmed reads to ensure adequate sequence quality, GC content, and removal of adapter sequences.

## Expression quantification

The reference transcriptome used in this study was created by combining the unique protein-coding transcripts from the Ensembl and NCBI RefSeq databases of the Rnor6.0 annotation, as previously described [23]. Initial analysis was conducted using the more up to date mRatBN7.2 reference transcriptome. However, it became apparent that many genes of particular interest were either missing (*Cd4*) or annotated as pseudogenes (*Lyve1*) in this new assembly. It was therefore decided to continue using the Rnor6 reference transcriptome for all analysis in this study.

After pre-processing, transcript expression level was quantified as transcripts per million using Kallisto (v0.46.0) [150]. Kallisto expression files were imported into RStudio (R version 4.2.1) using the tximport package (v1.24.0) to collate transcript-level TPM generated by Kallisto into gene-level TPMs for use by downstream tools.

## Differential expression analysis

Differential expression analysis was conducted with DESeq2 (v1.36.0) [151] with both 'condition' and 'sex' being used in the design model. An FDR cut-off of 0.05 was used, and only genes with a row sum of more than 10 raw counts were included. The volcano plots were created using the EnhancedVolcano package, and the lfcShrink function from DESeq2 was used with the 'apeglm' method to shrink log2 fold changes to reduce the effects of outliers.

## Network cluster analysis of gene expression

Gene expression network visualizations shown in Fig 1 were created using BioLayout (http://biolayout.org). All network cluster analysis was based on gene-level expression and was conducted using Graphia (https://graphia.app) [20]. Only genes expressed at ≥ 1 TPM in at least 2 samples were retained for analysis. BioLayout and Graphia calculate a Pearson correlation matrix comparing similarities in gene expression profiles between samples. For the global analysis, only relationships where r ≥ 0.8 were included. For all other analyses, only relationships where r ≥ 0.85 were included. A network graph was then created where nodes (genes) were connected by edges (representing correlation coefficients). Samples which exhibited similar patterns of gene expression across tissues were placed close together in the network graph. Gene expression patterns were further characterised using the Markov Cluster Algorithm (MCL) with an inflation value of 1.3 for global analysis and 2.0 for individual tissue analysis, to identify clusters of transcripts with similar expression patterns.

## Data visualisation

To provide an accessible method for visualising the transcriptomic rat atlas contained within this paper, a web app was created in R using the Shiny framework (https://shiny.posit.co/). The app is hosted using the shinyapps.io server at the following location (https://github.com/dylan-cartercusack/RNA-Seq-Web-App). The entire processed RNA-Seq dataset is also available for download as an Excel spreadsheet at the following location (https://doi.org/10.48610/5b57a22).

## Flow cytometry

**Tissue preparation.** Brain tissue for flow cytometry was collected from one hemisphere by performing a sagittal incision down the midline of the brain. This tissue was then minced with a scalpel in a petri dish, resuspended in HBSS containing collagenase type IV (1mg/mL), dispase (0.1mg/mL), and DNase (19mg/mL), and incubated on a rocking platform for 45 min at 37°C. The tissue was then passed through a 70μm filter, washed in 25mL PBS and centrifuged (at 400g and 4°C for 5 min). Supernatant was removed and the pellet resuspended in 12.5mL isotonic Percoll (2.44mL Percoll (Cytiva, MA, USA), 0.47mL 10x PBS, 7.815mL 1x PBS). The suspension was centrifuged at 800g for 45 min (no brake) at 4°C. The supernatant containing the top myelin layer was aspirated, and the pellet was washed twice with PBS, and finally resuspended in flow cytometry (FC) buffer (PBS/2% FCS).

Lung tissue for flow cytometry was prepared in the same way as brain, with the omission of the Percoll gradient.

**Standard flow cytometry.** Once a single cell suspension had been obtained, cells were resuspended in FC buffer for surface staining. $1 \times 10^6$ cells were stained for 30 minutes at 4°C in FC buffer containing unlabelled CD32 (BD Biosciences, NJ, USA) to prevent Fc receptor binding, HIS48–FITC, CD11B/C–BV510, CD45R–BV786, CD172–BV421, CD45–PE/Cy7 (BD Biosciences, NJ, USA), CD43–AF647 and CD4-APC/Cy7 (Biolegend, CA, USA). Cells were then washed twice, and finally resuspended in FC buffer containing 7AAD (Invitrogen, MA, USA). Flow cytometric acquisition was conducted on an LSRFortessa Cytometer (BD Biosciences, NJ, USA), and data was analysed using FlowJo v10 Software (BD Life Sciences, NJ, USA).

**C1Q ELISA.** Serum C1Q was measured using a recently developed quantitative sandwich ELISA [76]. An ELISA plate was coated with a monoclonal anti-C1Q capture antibody (9H10) for 1 hr at 37°C at 5μg/mL. Non-specific binding was blocked using 2% BSA in 0.05% PBS with Tween-20. Rat serum was added to the plate at a dilution of 1:1600 and incubated at 37°C for 2 hrs. The biotinylated monoclonal anti-C1Q detection antibody (2F6-biotin) was added for 1 hr at 37°C at 2ug/mL. The bound protein was detected using streptavidin-HRP. The

concentration of C1Q was calculated by regression of the standard curve generated using a 2-fold serial dilution series of purified rat serum C1Q as described [76].

**Histological staining.** Tissues were fixed in 4% paraformaldehyde for 24 hr. Standard paraffin processing and embedding methods were used (TRI Histology Core Facililty, QLD, Australia). 4μm (for H&E/elastin stains) and 10μm (for all other stains) sections were collected using a Leica Microtome (RM2245). Sections were deparaffinised and rehydrated in a descending ethanol series (3 x 100% xylene, 3 x 100% ethanol, 2 x 95% ethanol, 1 x 70% ethanol, RO $H_2O$). After staining, sections were dehydrated and mounted using DPX mountant (Sigma Aldrich, MO, USA). For H&E staining, sections were stained in Gill's No. 2 haematoxylin for 30 seconds and eosin for 40 seconds. For Sirius red staining, sections were stained in Picric-Sirius Red solution (Australian Biostain P/L, VIC, Australia) for 1hr. Cells from peritoneal lavages were fixed to slides with a cytospin, and then stained in Toluidine Blue working solution for 3 minutes. For elastin staining, 4 μm sections were stained with Elastin solution, Weigert's iron hematoxylin solution and Picrofuchsin solution for 10, 5 and 2 minutes respectively (Elastin van Gieson staining kit, Merck, Melbourne, Australia).

Antigen retrieval methods have been previously described [152]. CD209B, CD163 and Peripherin staining used method 2, while IBA1 used method 1. Antibody concentrations and staining times can be found in Table 2. Secondary detection was with DAKO Envision

**Table 2. Antibodies and Reagents.**

| Primary Antibodies | Target | Concentration | Incubation Time | Supplier | Catalogue Number |
|---|---|---|---|---|---|
| | CD163 | 1:500 | 60 min | Abcam | Ab182422 |
| | CD209B | 1:2000 | 60 min | Abcam | Ab308457 |
| | Peripherin | 1:10000 | 40 min | Abcam | Ab246502 |
| | IBA1 | 1:1000 | 120 min | Novachem | 019-19741 |
| **Flow cytometry antibodies** | **Target** | **Fluorochrome** | **Clone** | **Supplier** | **Catalogue Number** |
| | CD45 | PE-Cy7 | OX-1 | BD Biosciences | 561588 |
| | HIS48 | FITC | HIS48 | BD Biosciences | 554907 |
| | CD11B/C | BV510 | OX-42 | BD Biosciences | 743978 |
| | CD172 | AF405 | OX-41 | BD Biosciences | 744861 |
| | CD45R | BV785 | HIS24 | BD Biosciences | 743595 |
| | CD43 | AF647 | W3/13 | Biolegend | 202810 |
| | CD4 | APC-Cy7 | W3/25 | Biolegend | 201518 |
| | CD32 | NA | D34-485 | BD Biosciences | 550270 |
| | 7-Amino-actinomycin D | NA | NA | Invitrogen | A1310 |
| **Primer Target** | | **Forward Primer Sequence** | | **Reverse Primer Sequence** | |
| *Hprt* | | CTCAGTCCCAGCGTCGTGA | | AACACCTTTTC-CAAATCTTCAGCA | |
| *P2ry12* | | TTCCTGCTGTCACTGCCTAA | | TATCTCGTGCCAGACCAGAC | |
| *Sall1* | | AAGGCAATCTGAAGGTCCAC | | AACTTGACAGGATTCCCTCCT | |
| *Tmem119* | | AGAAGGGGAACAGGCCAGAG | | GCAGGCCCATCTGAATCAAC | |
| *Csf1r* | | GACTGGAGAGGAGAGAGCAGGAC | | GTGGAGGGCAGCAGGACTCT | |
| *Cd68* | | CTGTGACAGTGCCCATCCCC | | CTGTGGCTCTGATGTCGGTCC | |
| *Cd163* | | CAGGTGTTGTCTGCTCGGAGTT | | GCTGCCAATACTGCCCCATGT | |
| *C1qa* | | CGGGTCTCAAAGGAGAGAGA | | CAGATTCCCCCATGTCTCC | |
| *Ezh2* | | CAGCCTTGTGACAGTTCGTG | | GCATCCAGGAAAGCGGTTTT |
| *Akr1b1* | | AAGGAACCTGGTCGTGATCC | | TGTCCTCATTGCTCAGCTCA | |
| *Star* | | CGTGGCTGCTCAGTATTGAC | | AAGTGGCTGGCGAACTCTAT | |
| *Dbh* | | GAAGAATGCTGTGACTGTCCA | | TGGCCATTGTCCTGTTTTCTG | |

anti-rabbit HRP detection reagents (Agilent). Imaging was conducted on an Olympus V200 Slide Scanner.

Author Rachel Allavena, a board-certified veterinary anatomic pathologist, performed morphological identification of selected tissues including lung and diaphragm to confirm phenotypes.

**Statistics.** Statistical tests were performed using GraphPad Prism v9.4.1. Comparisons between WT and *Csf1rko* were conducted using unpaired, nonparametric Mann-Whitney tests. For comparisons of 3 or more, one-way ANOVA with multiple comparison testing was used.

## Supporting information

**S1 Fig. Pathology of *Csf1rko* lung.** Representative H&E and elastin images of 3-week-old WT and *Csf1rko* lung. Lungs were expanded and flushed with PBS via the trachea prior to fixation. * indicates examples of terminal bronchioles. Arrows point to regions of alveolar wall destruction, as defined by a board certified veterinary anatomic pathologist. Original magnification: 40X. Scale bar: 100μm.
(TIF)

**S2 Fig. Expression of IBA1 in lymphoid organs.** (A) Representative images showing immunohistochemical localization of CD209B (brown) in 3 week WT and *Csf1rko* liver. Scale bar: 100μm. (B) Representative images showing immunohistochemical localization of IBA1 (brown) in 3 week WT and *Csf1rko* thymus and mesenteric lymph node (mLN). Scale bar: 100μm.
(TIF)

**S3 Fig. The effect of *Csf1rko* on the diaphragm** . (A) Representative Masson's Trichrome images of 3 week WT and *Csf1rko* diaphragms. MD: muscular diaphragm, CT: central tendon. Muscular diaphragm region is showing the junction between muscular diaphragm and central tendon. Green arrows indicate regions of haemorrhage in both muscular diaphragm and the dense/loose connective tissue of the central tendon of *Csf1rko*. Scale bar: 100μm. (B) Gene set enrichment plot showing a reduction in collagen related genes in the *Csf1rko* diaphragm. Created using the C2 (curated gene sets) collection from the Molecular Signatures Database (MSigDB). NES – Normalised Enrichment Score; FDR – False Discovery Rate; positive enrichment score indicates an enrichment in the WT animals and a negative enrichment in the *Csf1rko* animals.
(TIF)

**S4 Fig. Repopulation of tissue macrophage populations in *Csf1rko* rats following BMT.** Representative whole-mount immunofluorescence images of tissues from *Csf1rko* rats 12 weeks after IP transfer of WT *Csf1r*-mApple donor bone marrow cells. No fluorescent signal is detected in any tissue in nontransgenic WT rats or in untransplanted *Csf1rko* rats. Scale bar: 200μm.
(TIF)

**S5 Fig. Repopulation of tissue macrophages in the *Csf1rko* following WT bone marrow cell transfer (BMT).** (A–C) Expression profiles in transcripts per million (TPM) for selected genes from RNA–Seq data of (A) peritoneal lavage (PT)(B) lung and (C) liver from WT and *Csf1rko* BMT recipients. Primary data is in S9 Table. Graphs show the mean ± SD. (D) qRT–PCR analysis of adrenal glands from WT and *Csf1rko* BMT recipients (10 WT; 9-10 *Csf1rko* + BMT). Graphs show the mean ± SD. *, P < 0.05; **, P < 0.01; ***, P < 0.001; ****, P < 0.0001.
(TIF)

**S6 Fig. Representative image of Toluidine Blue staining in peritoneal lavage cells from a 3 week old WT rat.** Scale bar: 100μm.
(TIF)

**S7 Fig. Pattern of repopulation of CD209b[+] macrophages in spleen following BMT** .
Representative images showing immunohistochemical localization of CD209B (brown) in *Csf1rko* BMT recipient spleens at different time points post BMT. 3 different regions of the same spleen are shown for 2 wks post BMT. Scale bar: 200μm. Original magnification for all images: 40X.
(TIF)

**S1 Table. Global Analysis.**
(XLSX)

**S2 Table. Lung.**
(XLSX)

**S3 Table. Lymphoid Organs.**
(XLSX)

**S4 Table. GI Tract.**
(XLSX)

**S5 Table. Muscle.**
(XLSX)

**S6 Table. Kidney and Adrenal Gland.**
(XLSX)

**S7 Table. Pituitary, Gonads and White Adipose Tissue.**
(XLSX)

**S8 Table. BMT Brain.**
(XLSX)

**S9 Table. BMT Other Organs.**
(XLSX)

**S10 Table. CSF1R Dependent Genes.**
(XLSX)

## Acknowledgments

We would like to thank the UQ PACE Biological Resources Facility staff for their assistance with breeding and husbandry of the *Csf1rko* rats. DAH/KMI/KMS gratefully acknowledge core laboratory support from The Mater Foundation. TRI is supported by the Australian Government.

## Author contributions

**Conceptualization:** Dylan Carter-Cusack, Kim M. Summers, Katharine M. Irvine, David A. Hume.

**Data curation:** Dylan Carter-Cusack.

**Formal analysis:** Dylan Carter-Cusack, Rachel Allavena, B. Paul Morgan.

**Funding acquisition:** David A. Hume.

**Investigation:** Dylan Carter-Cusack, Stephen Huang, Sahar Keshvari, Omkar Patkar, Anuj Sehgal, Rachel Allavena.

**Methodology:** Dylan Carter-Cusack, Stephen Huang, Sahar Keshvari, Robert A. J. Byrne, B. Paul Morgan, Stephen J. Bush, Kim M. Summers.

**Project administration:** David A. Hume.

**Supervision:** Stephen Huang, Kim M. Summers, Katharine M. Irvine, David A. Hume.

**Validation:** Dylan Carter-Cusack.

**Visualization:** Dylan Carter-Cusack.

**Writing – original draft:** Dylan Carter-Cusack, David A. Hume.

**Writing – review & editing:** Dylan Carter-Cusack, Rachel Allavena, B. Paul Morgan, Stephen J. Bush, Kim M. Summers, Katharine M. Irvine, David A. Hume.

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
