## [Decision Letter · Decision Letter 0]

5 Nov 2024

PGENETICS-D-24-01027Wild-type bone marrow cells repopulate tissue resident macrophages and reverse the impacts of homozygous CSF1R mutation.PLOS Genetics Dear Dr. Carter-Cusack, Thank you for submitting your manuscript to PLOS Genetics. After careful consideration, we feel that it has merit but does not fully meet PLOS Genetics's publication criteria as it currently stands. Therefore, we invite you to submit a revised version of the manuscript that addresses the points raised during the review process. Please submit your revised manuscript within 30 days Dec 05 2024 11:59PM. If you will need more time than this to complete your revisions, please reply to this message or contact the journal office at plosgenetics@plos.org. Please include the following items when submitting your revised manuscript:* A rebuttal letter that responds to each point raised by the editor and reviewer(s). You should upload this letter as a separate file labeled 'Response to Reviewers '. This file does not need to include responses to formatting updates and technical items listed in the 'Journal Requirements' section below.* A marked-up copy of your manuscript that highlights changes made to the original version. You should upload this as a separate file labeled 'Revised Manuscript with Track Changes '.* An unmarked version of your revised paper without tracked changes. You should upload this as a separate file labeled 'Manuscript '. If you would like to make changes to your financial disclosure, competing interests statement, or data availability statement, please make these updates within the submission form at the time of resubmission. Guidelines for resubmitting your figure files are available below the reviewer comments at the end of this letter. We look forward to receiving your revised manuscript. Kind regards, Kristin Leigh Patrick, Ph.D.Guest EditorPLOS Genetics Giovanni BoscoSection EditorPLOS Genetics Aimée DudleyEditor-in-ChiefPLOS Genetics Anne GorielyEditor-in-ChiefPLOS Genetics **Journal Requirements:** **Additional Editor Comments (if provided):** Reviewers were unanimous in commending the authors on the importance of this study and the impact of their findings. Some minor changes were suggested to improve the clarity of the figures and text; these should be incorporated before the manuscript can be considered for publication. Namely, as suggested by Reviewer 2, the manuscript would benefit from a more thoughtful analysis of some of the data and the authors' motivation for focusing on certain gene expression signatures. Where applicable, the authors should provide additional detail about the biological roles of the genes/clusters impacted by loss of CSF1R. Better labeling of histology images is also requested.**Reviewers' comments:** Reviewer's Responses to Questions

**Comments to the Authors:**

Reviewer #1: Carter-Cusack et al. provide a comprehensive, descriptive study on Csf1rko rats, both reconstituted with and without BM. It is encouraging to see that the macrophage population in rats resembles the gene transcript profiles of mice and humans. Among their findings, C1q emerged as a prominent gene signature, strongly associated with interstitial macrophages. Notably, interstitial macrophages do not develop in the absence of CSF1, as demonstrated in studies by Ural et al. (Sci Immunology, 2020) on the lung and Lim et al. (Immunity, 2018; 10.1016/j.immuni.2018.06.008) on the heart. Thus, the loss of gene signatures outlined in this study unmistakably points to interstitial macrophages, a population present across various organs and species. Recommended additional papers for assessing interstitial macrophage genes across various organs and species include the following:

Plantinga M, et al., Immunity 2013

Tamoutounour S et al., Immunity 2013

Epelman S, et al., Immunity 2014

Gibbings et al., AJRCMB 2017

Chakarov, et al., Science 2019

BB Ural et al., SciImmu 2020

Schyns, et al., NatCom 2019

Dick et al., SciImmu 2022

Xi Li, et al, Nature Imm 2024

Reviewer #2: The manuscript investigates the critical role of CSF1R-dependent macrophages in postnatal development using a rat model with a homozygous mutation in the Csf1r gene (Csf1rko). The authors demonstrate that the absence of these macrophages leads to significant growth retardation and organ developmental issues, which can be partially reversed through the transfer of wild-type bone marrow cells. They perform comprehensive RNA sequencing across various tissues, revealing distinct transcriptomic profiles, highlighting tissue-specific macrophage signatures and the systemic effects of macrophage deficiency. While the study provides valuable insights into the importance of macrophages for normal development and potential therapeutic strategies for CSF1R-related conditions, this manuscript is more descriptive and lacks causative validations in general. Areas for improvement include larger sample sizes, deeper mechanistic analysis, and a clearer discussion of the limitations of this study. Below are specific suggestions to enhance the manuscript:

1. While the analysis demonstrates CSF1R dependency across tissues, particularly non-lymphoid organs, Figure 1 could benefit from additional annotations that highlight critical findings. For instance, identifying clusters with substantial CSF1R-dependent gene expression changes would strengthen the figure's biological relevance, especially regarding key clusters such as the macrophage-associated genes found in subsequent analyses.

2. The immunohistochemistry images in part (E) of Figure 3 are labeled, but it could be more informative if the text explicitly explained the cellular localization or morphological distinctions shown between WT and KO groups. The authors should put arrows on the Figure 3E to indicate the WT and KO differences.

3. Figure 4 demonstrates the immunohistochemical localization of CD209B in the thymus, spleen, and mesenteric lymph nodes at 40X magnification with a 100µm scale bar, yet it lacks clarity on notable qualitative or quantitative differences across tissues, which would help distinguish each representation. Additionally, the volcano plot showing differential expression (DE) between WT and Csf1rko lymph nodes, calculated using DESeq2 with log fold changes adjusted via lfcShrink, could benefit from more detailed context on specific genes or clusters impacted by this DE analysis to enhance understanding of key findings.

4. Figure 5 discusses gene expression profiles in WT and Csf1rko samples across gut tissues (colon, ileum, and duodenum) but lacks clarity on gene selection criteria and specific affected clusters, which would aid comprehension. It groups Csf1r-dependent genes into those common across all gut tissues, those more impacted in the colon and duodenum, and DC-associated genes elevated in the Csf1rko ileum, though further explanation of this segmentation’s relevance to the study is needed. Additionally, Figure 5D’s immunohistochemical staining for peripherin in the ileum and its quantification require more context to clarify its role in neuronal development or gut function within the Csf1rko model.

5. In Figure 6, the authors fails to provide sufficient context or clarity regarding specific findings. Details about the methods used and the significance of the results, such as whether the observations align with prior data, are minimal. Additionally, explanations connecting visual elements (like particular markers or staining patterns) to study conclusions are sparse, making it difficult for readers to understand the implications. Including these contextual details would enhance the text’s clarity and strengthen the link between visual data and the conclusions drawn.

6. For Figure 7, the main issues in the manuscript include a lack of clarity in the rationale for focusing on specific gene expression profiles in the kidney and adrenal gland. Additionally, there is insufficient contextual information about the functional relevance of selected genes, particularly regarding their potential impact on the organ development being studied. The calcium concentration measurement results would benefit from more detailed interpretation to strengthen the findings. The imaging details, especially for the adrenal gland, could also be improved to provide a more comprehensive understanding of the histological differences between WT and Csf1rko samples. This would enhance reader comprehension of both the structural and molecular alterations under investigation.

7. The main concerns regarding the descriptions for Figure 8 focus on the clarity and depth of the gene expression analysis across multiple tissues. There is a need to specify better the rationale behind selecting particular genes for expression profiling in WT and Csf1rko samples and to discuss the impact of differentially expressed genes within the intestinal regions examined. Additionally, more explicit detail on tissue-specific gene clusters and their significance would enhance the figure’s interpretive value.

Reviewer #3: I would like to thank the authors for their valuable insights into the role of macrophages in rats with a Csf1r mutation, which reveal that the absence of these cells impairs growth and organ maturation. The bone marrow transplant successfully rescues macrophage populations and normalizes function, reversing the associated pathology. These findings enhance our understanding of CSF1R mutations and the innate immune deficiencies related to prematurity.

All figures are self-explanatory, and the manuscript is well-written. Based on this, I recommend accepting the manuscript for publication with minor revisions.

Major:

It would be helpful to discuss why circulating C1Q is undetectable in Csf1rko rats.

While I understand the study is comprehensive, I recommend condensing the discussion section to focus on the main findings.

Minor:

Please define Csf1r in both the abstract and the main text.

**Have all data underlying the figures and results presented in the manuscript been provided?**

Reviewer #1: Yes

Reviewer #2: None

Reviewer #3: Yes

PLOS authors have the option to publish the peer review history of their article (what does this mean? ). If published, this will include your full peer review and any attached files.

**Do you want your identity to be public for this peer review?** For information about this choice, including consent withdrawal, please see our Privacy Policy .

Reviewer #1: No

Reviewer #2: No

Reviewer #3: **Yes: ** GAJANAN DATTATRAY KATKAR

 **Figure resubmission:** While revising your submission, please upload your figure files to the Preflight Analysis and Conversion Engine (PACE) digital diagnostic tool, https://pacev2.apexcovantage.com/. PACE helps ensure that figures meet PLOS requirements. To use PACE, you must first register as a user. Registration is free. Then, login and navigate to the UPLOAD tab, where you will find detailed instructions on how to use the tool. If you encounter any issues or have any questions when using PACE, please email PLOS at figures@plos.org. Please note that Supporting Information files do not need this step. If there are other versions of figure files still present in your submission file inventory at resubmission, please replace them with the PACE-processed versions. **Reproducibility:** To enhance the reproducibility of your results, we recommend that authors deposit laboratory protocols in protocols.io, where a protocol can be assigned its own identifier (DOI) such that it can be cited independently in the future. Additionally, PLOS ONE offers an option to publish peer-reviewed clinical study protocols. Read more information on sharing protocols at https://plos.org/protocols?utm_medium=editorial-email&utm_source=authorletters&utm_campaign=protocols

---

## [Editor Report · Decision Letter 1]

4 Dec 2024

Dear Dr Carter-Cusack,

We are pleased to inform you that your manuscript entitled "Wild-type bone marrow cells repopulate tissue resident macrophages and reverse the impacts of homozygous CSF1R mutation." has been editorially accepted for publication in PLOS Genetics. Congratulations!

Yours sincerely,

Kristin Leigh Patrick, Ph.D.

Guest Editor

PLOS Genetics

Giovanni Bosco

Section Editor

PLOS Genetics

Aimée Dudley

Editor-in-Chief

PLOS Genetics

Anne Goriely

Editor-in-Chief

PLOS Genetics

Comments from the reviewers (if applicable):

We think the authors have gone above and beyond in responding to reviewers comments. The manuscript will stand as an important resource for the immunology community at large.

**Data Deposition**

http://datadryad.org/submit?journalID=pgenetics&manu=PGENETICS-D-24-01027R1

**Press Queries**

---

## [Editor Report · Acceptance letter]

PGENETICS-D-24-01027R1

Wild-type bone marrow cells repopulate tissue resident macrophages and reverse the impacts of homozygous CSF1R mutation.

Dear Dr Carter-Cusack,

We are pleased to inform you that your manuscript entitled "Wild-type bone marrow cells repopulate tissue resident macrophages and reverse the impacts of homozygous CSF1R mutation." has been formally accepted for publication in PLOS Genetics! Your manuscript is now with our production department and you will be notified of the publication date in due course.

With kind regards,

Anita Estes

PLOS Genetics

On behalf of:
